# Running Vacuum in the Universe: Phenomenological Status in Light of the Latest Observations, and Its Impact on the $\sigma_8$ and $H_0$ Tensions

Joan Solà Peracaula [1,*] , Adrià Gómez-Valent [2,3] , Javier de Cruz Pérez [4] and Cristian Moreno-Pulido [1]

1   Departament de Física Quàntica i Astrofísica, and Institute of Cosmos Sciences, Universitat de Barcelona, Av. Diagonal 647, 08028 Barcelona, Spain; cristian.moreno@fqa.ub.edu
2   Istituto Nazionale di Fisica Nucleare, Sezione di Roma 2, Via della Ricerca Scientifica 1, 00133 Roma, Italy; agvalent@roma2.infn.it
3   Dipartimento di Fisica, Università degli Studi di Roma Tor Vergata, Via della Ricerca Scientifica 1, 00133 Roma, Italy
4   Departamento de Física Teórica, Universidad Complutense de Madrid, 28040 Madrid, Spain; jadecruz@ucm.es
*   Correspondence: sola@fqa.ub.edu

**Abstract:** A substantial body of phenomenological and theoretical work over the last few years strengthens the possibility that the vacuum energy density (VED) of the universe is dynamical, and in particular that it adopts the 'running vacuum model' (RVM) form, in which the VED evolves mildly as $\delta\rho_{\rm vac}(H) \sim \nu_{\rm eff} m_{\rm Pl}^2 \mathcal{O}(H^2)$, where $H$ is the Hubble rate and $\nu_{\rm eff}$ is a (small) free parameter. This dynamical scenario is grounded on recent studies of quantum field theory (QFT) in curved spacetime and also on string theory. It turns out that what we call the 'cosmological constant', $\Lambda$, is no longer a rigid parameter but the nearly sustained value of $8\pi G(H)\rho_{\rm vac}(H)$ around any given epoch $H(t)$, where $G(H)$ is the gravitational coupling, which can also be very mildly running (logarithmically). Of particular interest is the possibility suggested in past works that such a running may help to cure the cosmological tensions afflicting the $\Lambda$CDM. In the current study, we reanalyze the RVM in full and we find it becomes further buttressed. Using modern cosmological data, namely a compilation of the latest SNIa+BAO+$H(z)$+LSS+CMB observations, we probe to what extent the RVM provides a quality fit better than the concordance $\Lambda$CDM model, with particular emphasis on its impact on the $\sigma_8$ and $H_0$ tensions. We utilize the Einstein–Boltzmann system solver `CLASS` and the Monte Carlo sampler `MontePython` for the statistical analysis, as well as the statistical DIC criterion to compare the running vacuum against the rigid vacuum ($\nu_{\rm eff} = 0$). On fundamental grounds, $\nu_{\rm eff}$ receives contributions from all the quantized matter fields in FLRW spacetime. We show that with a tiny amount of vacuum dynamics ($\nu_{\rm eff} \ll 1$) the global fit can improve significantly with respect to the $\Lambda$CDM and the mentioned tensions may subside to inconspicuous levels.

**Keywords:** cosmology; cosmological constant; vacuum energy; dark energy; quantum field theory

## 1. Introduction

The vanilla concordance model of cosmology, or standard $\Lambda$CDM model (the current standard model of cosmology with flat three-dimensional geometry), is based on the Friedmann–Lemaître–Robertson–Walker (FLRW) metric and has been a rather successful paradigm for the phenomenological description of the universe for more than three decades [1,2]. Its consolidation after a solid observational underpinning, however, was only possible in the late nineties [3]. The vanilla model has remained robust and unbeaten for a long time, as it is essentially consistent with a large body of observations. These have indeed provided strong support for a spatially flat and accelerating universe in the present time. The ultimate cause of such an acceleration is currently unknown, but it is attributed to an energy component in the universe popularly called "dark energy" (DE), which may

adopt a large number of picturesque forms depending on the favored theoretical preference of different cosmologists, see, e.g., [4] for a large variety of options. DE constitutes ∼70% of the total energy density of the universe and presumably possesses enough negative pressure as to produce the observed cosmic acceleration. Nevertheless, the nature of DE remains a complete mystery. The simplest candidate is the cosmological term in Einstein's equations, $\Lambda$, usually assumed to be constant, which is why it is usually called the cosmological constant (CC) [5,6]. Consistent observational measurements of $\Lambda$ (treating it as a mere fit parameter) made independently in the last quarter of a century using distant type Ia supernovae (SNIa), the baryonic acoustic oscillations (BAO), and the anisotropies of the cosmic microwave background (CMB), have formed the foundations of the concordance $\Lambda$CDM model of cosmology [7–15].

Despite the vanilla ($\Lambda$CDM) model faring relatively well with the current observational data, it traditionally suffers from a variety of problems of different kinds which seriously challenge its credibility. For a long time, people somehow decided to turn a blind eye to the deepest questions and also to different spots and wrinkles which perturb that flawless and immaculate condition. The profound theoretical problems (and the practical wrinkles as well) are nonetheless still there, alive and kicking, whether we wish to look at them or not. First and foremost, the hypothetical existence of dark matter (DM) still lacks direct observational evidence. On a deeper level of mystery, the nature and origin of DE (the dominant component of the cosmic energy budget) still lies in the limbo of the most unfathomable cosmological riddles. Because if we admit the simplest proposal for DE, that is to say, the cosmological constant $\Lambda$, one has to cope with the 'cosmological constant problem' [16], perhaps the most inscrutable problem ever in theoretical physics and cosmology [17]. It manifests itself in a dual manner, to wit: the fine-tuning problem associated with the large value of $\Lambda$ predicted by most theoretical approaches ("the old CC problem" [16]); and also what it has become customary to call the 'cosmic coincidence problem' [18]; see also [17,19,20] for a discussion of these enigmas, which lie at the interface between cosmology and quantum field theory (QFT).

The toughest conundrum of all is probably that of explaining the relation between $\Lambda$ and the vacuum energy density (VED): $\rho_{\rm vac} = \Lambda/(8\pi G_N)$, where $G_N$ is Newton's constant. Traditionally one assumes that the corresponding pressure is $p_{\rm vac} = -\rho_{\rm vac}$. In this respect, we note that in 1934 G. Lemaître pointed out the following [21]: "Everything happens as though the energy in vacuum would be different from zero. In order that it shall not be possible to measure motion relative to the vacuum, we have to associate a pressure to the energy density of the vacuum. This is essentially the meaning of the cosmological constant."; see also [22]. The interrelationship between VED and $\Lambda$ in the general quantum theory context has been assessed by theoretical physicists for more than a century, as in the days of W. Nernst and W. Pauli. At that time the issue was already troublesome [17]. However, the most severe implications in the cosmological arena took shape only with the development of the formal aspects of QFT. It is in this modern theoretical context where the notion of VED seems to cause a serious conflict with the cosmological measurements, the reason being that the typical contribution from the vacuum fluctuations of any quantum field of mass $m$ is expected (on mere dimensional grounds) to be proportional to the quartic power of its mass: $\rho_{\rm vac} \propto m^4$, as noted by Zeldovich [23,24]. Such a prediction must be compared with the measured value of $\Lambda$ expressed in terms of the corresponding VED, which is $\rho_{\rm vac} \sim 10^{-47}$ GeV$^4$ in natural units. This is extremely small in comparison with the energy density that one may estimate using any particle physics mass $m$ from, say, the electronvolt scale to the mass scale of the weak gauge bosons in electroweak theory, $W^{\pm}$ and $Z$ (∼80, 90 GeV), the Higgs mass (∼125 GeV) and the top quark mass (∼170 GeV). The exception would be, of course, a millielectronvolt neutrino [17], but for any typical standard model particle the value of $\rho_{\rm vac} \propto m^4$ is mind-bogglingly too large, being indeed dozens of orders of magnitude larger than the measured value of $\Lambda$, not to speak of the situation in the grand unified theories (GUT's), where the characteristic energy scale can reach ∼$10^{16}$ GeV. It is because of the cosmological constant problem phrased on these

grounds that the VED option became outcast, as if it were to be blamed for all evils. The aforesaid notwithstanding, the criticisms usually have nothing better to offer, except to defend tooth and nail particular forms of the DE without providing any explanation about the genuine subject involved in the original discussion of this problem, which is, of course, to understand the role played by the VED in QFT and its fundamental relation with $\Lambda$. Most, if not all, proposed forms of DE, are actually plagued with the same (purported) fine-tuning problem that is attributed (in a way by fiat) to the vacuum option exclusively. This is certainly the case with, e.g., the popular family of quintessence models, phantom fields, and generalizations thereof; see, e.g., [5,6,25,26] and references therein. Attempts to understand the vacuum energy as a form of repulsive gravitation capable of driving the slow accelerating expansion of the universe, notwithstanding its exceedingly large value, have been presented in the literature; see, e.g., [22,27,28] for a review.

In recent years, new approaches to the notion of vacuum energy in QFT and its relation with the $\Lambda$-term suggest that these problems can be smoothed out to a large extent. In fact, the VED can be properly renormalized in QFT in curved spacetime, thereby offering a tamer theoretical context for the traditional vacuum energy approach to fit in with the observations. In light of these developments, the quantum vacuum energy could well be after all the most fundamental explanation for the DE in our universe; see, e.g., [17,19], as well as the latest formal developments in [29–32], summarized in [20].

The vanilla $\Lambda$CDM model, to which modern cosmological observations have converged in the last decades, is certainly an important triumph in our description of the main background features of the cosmic expansion and the large-scale structure formation processes in the universe. However, it is only a partial success. Its exceeding simplicity eventually turned into a perilous double edged sword; in fact, the absence of any connection with fundamental physics is the literal expression of such a simplicity and is most likely at the root of its many shortcomings. In truth, the $\Lambda$CDM does not possess enough theoretical structure to explain the success of the observations (e.g., the measured value of $\Lambda$) in a fundamental context, and at the same time it cannot even provide an explanation for other measurements that are threatening its viability. If we pay attention to the existing conflicts on several active fronts, the observational situation of the $\Lambda$CDM in the last decade or so does not seem to paint a fully rosy picture anymore. Beyond formal theoretical issues, a series of practical problems of a more mundane nature than those mentioned above are piling up as well [33]. On a mere phenomenological perspective, the situation is particularly worrisome, with some "tensions" existing with the data. For example, it has long been known that there appear to exist potentially serious discrepancies between the CMB observations (based on the vanilla $\Lambda$CDM) and the local direct (distance ladder) measurements of the Hubble parameter today [34]. The persisting mismatch between these measurements is what has been called the "$H_0$ tension". It is arguably the most puzzling open question within the current cosmological paradigm and it leads, if taken at face value, to a severe discrepancy of $\sim 5\sigma$ c.l. or more between the mentioned observables. Many proposals have been put forward to shed some spark of light onto that puzzling cosmological imbalance. Among the possibilities debated in the literature, it has been conjectured, e.g., that it could stem from a possible intrinsic "running of $H_0(z)$ with the redshift", presumably connected with the differences that may appear in the (total) effective equation of state (EoS) of the universe between the vanilla cosmology and the actual FLRW model underlying the observations [35,36]. While these are interesting possibilities, we are probably still far away from the resolution of this conundrum on fundamental grounds. At the same time, there exists a smaller, but appreciable ($\sim 2 - 3, \sigma$) tension in the realm of the large-scale structure (LSS) growth data, called the "$\sigma_8$ tension" [37]. This is concerned with the measurements of weak gravitational lensing at low redshifts ($z < 1$). Such a tension is usually evaluated with the help of the parameter $S_8$ or, alternatively, by means of $\sigma_8$; recall that $S_8 \equiv \sigma_8 \sqrt{\Omega_m^0/0.3}$. It turns out that these measurements favor matter clustering weaker than that expected from the vanilla model using parameters determined by CMB measurements; see, e.g., [38–45]. Recently, it has been claimed that $S_8$ values determined from

$f\sigma_8$ increase with redshift in the $\Lambda$CDM [46], which, according to these authors provides additional support to the fact that such a discrepancy may be physical in origin and with a value in the enhanced $2 - 4, \sigma$ range. In the constant pursuit of a possible late-time solution to these tensions, it has been argued that within the large class of models where the DE is treated as a fluid with EoS $w(z)$, solving the $H_0$ tension demands the phantom condition $w(z) < -1$ at some $z$, while solving both the $H_0$ and $\sigma_8$ tensions requires $w(z)$ to cross the phantom divide and/or other sorts of transitions; see, e.g., [47–52]. Specific realizations of the noticed double condition for the DE fluid can be found in the literature, e.g., in the context of the $\Lambda$XCDM model [53,54], closely related to the idea of the running vacuum to be discussed in the present work, see below. For detailed reviews on these tensions and other challenges afflicting the concordance $\Lambda$CDM model; see, e.g., [33,55,56] and the long list of references quoted therein bearing relation to these matters.

The severity of some of these tensions, and the huge number of proposals existing in the literature trying to explain them through a large disparity of ideas, suggests that it is perhaps time to come to grips anew with the fundamentals of the theoretical formulations, such as quantum field theory and string theory. We have already pointed to recent calculations claiming a more adequate renormalization prescription for the VED in quantum field theory in FLRW spacetime, leading to the "running vacuum model" (RVM). It turns out that this QFT approach may have a real impact not only on the more formal theoretical problems described in the beginning, but also on the practical issues concerning the aforementioned tensions. In fact, the VED resulting from the RVM leads to a time-varying vacuum energy density, and hence a time-varying (physical) $\Lambda$ as well, in which $\Lambda$ acquires a dynamical component through the quantum vacuum effects: $\Lambda \to \Lambda + \delta\Lambda$. The shift $\delta\Lambda$ is calculable in QFT since it depends on the contributions from the quantized matter fields (bosons and fermions). Upon appropriate renormalization, one finds that $\delta\Lambda$ depends on a term of the order of the Hubble rate $H(t)$ squared [29]: $\delta\Lambda \sim \nu, \mathcal{O}(H^2)$ ($\nu \ll 1$). This is the typical form of the RVM. The connection of the latter with QFT can be motivated from semi-qualitative renormalization group arguments on scale-dependence; see the reviews [17,19]. In particular, we mention the old works [57,58] and also recent approaches along these lines, such as [59]. However, an explicit QFT calculation leading to that form of 'running $\Lambda$' (associated with the 'running VED') appeared only very recently [29–32].

A note of caution is in order here. Over time, a large variety of cosmological models have been proposed to describe the DE and its possible dynamics. Apart from the aforementioned quintessence and similar models [5,6,25,26], there is a very populated habitat of models with a generic time-dependent cosmological constant, the so-called "$\Lambda(t)$-cosmologies". Many of these models, however, are of a purely phenomenological nature, since the time dependence of $\Lambda(t)$ is parameterized in an ad hoc manner. They might have a connection with fundamental theory, but it is not implemented in an explicit way in the corresponding papers. The list of models of this type is large and we will cite here only a few of them [60–79]; see also the old review [80]. In some cases, the parameterization is performed through a direct function of the cosmic time or of the scale factor, and sometimes as a function of the Hubble parameter, or even a hybrid combination of these various possibilities. Be that as it may, the general and rather nonspecific class of the "$\Lambda(t)$-cosmologies" should not be confused with the "running vacuum models" (RVMs) discussed above, in which the running of $\Lambda$ stems from the quantum effects on the effective action of QFT in curved (FLRW) spacetime. In other words, the RVMs are to be understood in a much more restricted sense; in fact, one that is closer to fundamental aspects of QFT, and only this precise type of time-evolving VED cosmology will be dealt with here. Let us finally note that, apart from the QFT formulation, a 'stringy' version of the RVMs is also available which can be very promising too [81–84]. The potential dynamics of the cosmic vacuum is, therefore, well motivated from different theoretical perspectives, and this fact further enhances the interest for

the current study, whose main purpose is to focus exclusively on the phenomenological implications of the class of RVM models.

We should mention that the running vacuum framework has already been tested with considerable success in previous works over the years. It has been known for quite some time that the RVM-type of cosmological models can help in improving the overall fit to the cosmological observations and also in smoothing out the mentioned tensions as compared to the ΛCDM; see, for instance, [41,85–95] for a short summary. For this reason, we believe it is worthwhile to keep on exploring the phenomenological consequences of the running vacuum in the light of the latest observations on all of the main data sources: SNIa+BAO+$H(z)$+LSS+CMB. The state-of-the-art-phenomenological performance of RVMs was reported not too long ago, in [85]. In the current work, however, we definitely enhance the scope of the results presented in that paper by considering an updated cosmological dataset in combination with an extended analysis of the CMB part. In point of fact, the main focus in this paper is to delve into the practical ability of the RVM to tackle the $\sigma_8$ and $H_0$ tensions versus the vanilla ΛCDM model. It is reassuring to find that the global fit to the cosmological observations can be improved within the running vacuum framework with respect to the ΛCDM. The optimal situation is when the VED presents a threshold in the recent past, where its dynamics becomes activated, and/or when the gravitational coupling is mildly running.

All in all, the dynamical DE models may offer a clue not only to relieve some high-level aspects of the cosmological constant and coincidence problems, but also to straighten out some very practical ones, such as helping to modulate the processes of structure formation which may impinge positively on the $\sigma_8$ tension. Last but not least, they can help to explain the existing mismatch between the distinct values of $H_0$ derived from measurements of the local and the early universe.

The paper is organized as follows. In Section 2, we present the running vacuum model (RVM) from a phenomenological point of view and emphasize its connection with QFT in curved spacetime. For convenience, we introduce the model variant of the RVM which we call RRVM, as we did in [85]. In it, the VED can be expressed entirely in terms of the curvature scalar $\mathcal{R}$ (which is of order $\sim H^2$) at the background level. We study two types of RRVMs: type I and type II. The type depends on whether the gravitational coupling $G$ is fixed at its current local gravity value, $G = G_N$, or evolving mildly with the expansion, $G = G(H)$, a feature which in our case is linked to the interaction or not, respectively, of the evolving vacuum energy with cold dark matter (CDM). Type I is studied at length in Section 3, where we describe the background cosmological equations and their solutions under appropriate conditions. At the same time, we discuss the corresponding perturbation equations. Type II, on the other hand, is studied in detail in Section 4, where again we provide the background solution and analyze the perturbations. In Section 5, we enumerate and briefly describe the different sources of observational data employed in this paper and the methodology used to constrain the free parameters of the models under discussion. We also define the four characteristic datasets (Baseline and Baseline+SH0ES with and without CMB polarization data) that will be used to test the running vacuum models and their comparison with the vanilla ΛCDM model. The outcome of our analyses under the different datasets is presented and discussed in detail in Section 6. Finally, in Section 7, we summarize our findings (see Tables 4–7 and Figures 1–4) and present the main conclusions of this study. In Appendix A, at the end of our work, we include additional tables with a detailed breakdown of the different $\chi^2$ contributions from each observable.

## 2. Running Vacuum in the Universe

As indicated, throughout our study we will assume that the background spacetime is FLRW with flat three-dimensional hypersurfaces. The general low-energy form of the vacuum energy density (VED) within the running vacuum model (RVM) has been explored phenomenologically on several previous occasions and with a remarkable degree of success, in the sense that in all cases it has proven to be rather competitive with the ΛCDM and

even able to surpass the fitting performance of the latter; see, e.g., [41,85–93]. Herein, we shall test if this is still the case with the current wealth of observations and using the state-of-the-art methods of analysis of the cosmological data. The dynamical structure of the running VED adopts the perspicuous form [29,30]

$$\rho_{\text{vac}}(H) = \frac{3}{8\pi G_N}\left(c_0 + \nu H^2 + \tilde{\nu}\dot{H}\right) + \mathcal{O}(H^4). \tag{1}$$

For $\nu = \tilde{\nu} = 0$, this expression reduces to $\rho_{\text{vac}} = \Lambda_{\text{phys}}/(8\pi G_N)$, where $\Lambda_{\text{phys}} = 3c_0$ retakes the traditional role of the cosmological constant term. However, for nonvanishing values of the coefficients $\nu$ and $\tilde{\nu}$, the vacuum acquires a certain amount of dynamics, which is provided by the $H^2$ and $\dot{H}$ contributions. Here, the dot indicates the derivative with respect to the cosmic time and $H = \dot{a}/a$ is the Hubble function. As we can see, the two leading dynamical terms of $\rho_{\text{vac}}$ in Equation (1) are both of $\mathcal{O}(H^2)$ since $\dot{H} \sim H^2$, this being true both in the matter- and radiation-dominated epochs. Despite the fact that the higher-order powers $\mathcal{O}(H^4)$ in the above expression are also predicted in the QFT context along with the lower-order ones $\mathcal{O}(H^2)$ [30], the former are unimportant for the current universe and will be hereafter ignored in favor of the latter. The additive parameter $c_0$ is constrained to satisfy $\rho_{\text{vac}}(H_0) = \rho_{\text{vac}}^0$, where $\rho_{\text{vac}}^0$ is the value of the VED today, and hence is connected with the physical value of the measured cosmological constant through $\rho_{\text{vac}}^0 = \Lambda_{\text{phys}}/(8\pi G_N)$. In this case, however, $\Lambda_{\text{phys}} \neq 3c_0$ is a quantity nontrivially connected to the dynamical terms in Equation (1), since a formal renormalization of the theory becomes necessary within the QFT context [29,30]. Upon renormalization, the bulk of the physical value is still provided by $c_0$ since the two (dimensionless) coefficients $\nu$ and $\tilde{\nu}$ adjoined to the two dynamical terms in (1) are expected to be small ($\nu, \tilde{\nu} \ll 1$) [96]. They encode the running character of the vacuum at low energy and can be computed in QFT in curved spacetime, receiving contributions from the quantized bosons and fermion fields. The explicit calculation was first presented in [29,30] and was recently completed in [32]. In these references, it is shown that the above VED structure can be formally derived from quantum effects on the effective action of QFT in FLRW spacetime.

Indeed, as shown in the works [29–32], fully within the spirit of the renormalization group (RG) analysis inherent to the RVM structure [17], the Hubble rate $H$ (with the natural dimension of energy) can be viewed as an RG scaling parameter. For the sake of simplicity, let us illustrate the type of effects that contribute to the coefficient $\nu$ of the $H^2$ terms in the VED (1), albeit focusing only on one quantized scalar field of mass $m$ and non-minimal coupling $\zeta$. The independent contributions to $\tilde{\nu}$ will not be shown [30] for this summarized discussion of the QFT aspects, and will be assumed to vanish. Then, one may write for the RVM energy density $\rho_{\text{vac}}(H)$ which connects two given values of the Hubble parameter, say, one at the $H$ era and the other at another epoch (the current one, $H_0$, for example):

$$\rho_{\text{vac}}(H) = \rho_{\text{vac}}^0 + \frac{3\nu}{8\pi G_N}\left(H^2 - H_0^2\right), \tag{2}$$

where

$$\nu = \frac{1}{2\pi}\left(\zeta - \frac{1}{6}\right)\frac{m^2}{m_{\text{Pl}}^2}\ln\left(\frac{m^2}{H_0^2}\right), \tag{3}$$

in which $m_{\text{Pl}}$ is the Planck mass. According to the QFT calculation, $\nu$ actually appears as a very mildly (logarithmically) dependent function of $H = H(t)$—see the exact expression in [30]. Since it remains essentially invariable with $H$, for fitting purposes we can fix $H = H_0$ in it and take $\nu$ as a constant fitting parameter. Notice also that in arriving at Equation (2), $\ln(m^2/H_0^2) \gg 1$ has been used (an inequality which is valid for virtually any massive particle). The previous considerations justify in part the structure of the VED in Equation (1). Regarding the $\dot{H}$ term, it has a similar structure [30], but the above considerations should suffice to grasp the kind of QFT contributions that are found. As

previously noted, $\zeta$ is the non-minimal coupling of the (quantized) scalar matter fields with gravity (in general, one expects a different coupling for each scalar field). In the conformal limit (certainly not our case) one would have $\zeta = 1/6$ and the running of the VED from the scalar sector would disappear, as could be expected. The quantity $\rho_{\text{vac}}^0$ in (2) denotes the vacuum energy density at the current era and hence it is connected with the measured value of the cosmological 'constant' through the aforementioned relation $\rho_{\text{vac}}^0 = \Lambda_{\text{phys}}/(8\pi G_N)$.

We should also mention, for the sake of a more complete summary of this QFT part, that the exact formula for scalars (including an arbitrary number of them) can be found in [30], and that the calculation of the scalar contribution recently became complemented with the QFT calculation of the (quantized) fermionic contributions, also in an arbitrary number; see [32]. It is therefore clear, that at present a theoretical quantitative prediction for the coefficients $\nu$ and $\tilde{\nu}$ cannot be performed in practice since they depend on contributions from the masses and non-minimal couplings of the various scalar fields, as well as from the masses of the different species of fermions. Furthermore, from the above formulas it should be clear that the relevant fields contributing significantly (at one-loop and higher orders) to the VED running are not the low-energy fields of the standard model of particle physics but the very heavy fields belonging to the given grand unified theory (usually accompanied with large multiplicity factors). All that being said, and despite the fact that a quantitative QFT prediction of the cosmic running of the VED cannot be presently furnished, the great virtue of these formal calculations is, at least from our point of view, that they provide a theoretical link between cosmologically relevant quantities and the QFT framework, and hence contribute to establishing a deeper connection of cosmology with the fundamental principles of theoretical physics.

In practice, therefore, the values of $\nu$ and $\tilde{\nu}$ must be fitted to the cosmological observations. Thus, in what follows we will focus exclusively on the phenomenological consequences of the RVM. What is important is that these coefficients are expected to be small and of order $\sim M_X^2/m_{\text{Pl}}^2 \ll 1$, where $m_{\text{Pl}} \simeq 1.22 \times 10^{19}$ GeV is the Planck mass and $M_X \sim 10^{16-17}$ GeV is of the order of a typical GUT scale (or even a string scale slightly above it) times a multiplicity factor accounting for the number of heavy particles in the GUT [96]. For $\nu = \tilde{\nu} = 0$ we recover the $\Lambda$CDM smoothly. This is a very welcome property of the RVM since DE models having no smooth $\Lambda$CDM limit, e.g., predicting a VED of the form $\rho_{\text{vac}}(H) \propto H^2$ or a combination of $H^2$ and $\dot{H}$ (without any additive term), would be excluded owing to their absence of an inflexion point from deceleration into acceleration in the cosmic evolution; see [93,97,98]. The presence of the nonvanishing additive term $c_0$ is therefore crucial for the RVM to avoid this unwanted situation, something that other models (e.g., entropic and ghost models of the DE) cannot avoid, thereby getting into trouble [99,100]. Holographic models with dynamical cutoff $L = H^{-1}$ also lack an additive term in the DE and are also unfavored already at a purely cosmographic level [101]. In stark contrast, the condition $c_0 \neq 0$ is always warranted within the class of RVMs.

It is important to realize that the dynamics of the VED must preserve, of course, the Bianchi identity satisfied by the Einstein tensor. In practice, this means that the total energy–momentum tensor (EMT), which receives the contributions from nonrelativistic matter, radiation, and vacuum (assumed here to be ideal fluids), must be covariantly conserved, namely $\nabla^\mu T_{\mu\nu}^{\text{tot}} = 0$. The total EMT reads,

$$T_{\mu\nu}^{\text{tot}} = (p_t - \rho_{\text{vac}})g_{\mu\nu} + (\rho_t + p_t)U_\mu U_\nu, \tag{4}$$

where $U^\mu$ is the 4-velocity vector of the cosmic fluid. We have defined $\rho_t = \rho_m + \rho_{\text{ncdm}} + \rho_\gamma$, where $\rho_m = \rho_{\text{cdm}} + \rho_b$ denotes the contribution to the proper density of nonrelativistic matter from cold dark matter and baryons, $\rho_{\text{ncdm}}$ (non-cold dark matter) corresponds to the energy density of neutrinos, and $\rho_\gamma$ designates the energy density associated with photons. In other words, $\rho_t$ refers to the sum of all the species in the universe excluding the vacuum. Analogous notation applies to the pressures. We shall, however, be more specific in our treatment of the various contributions to the EMT in the next section. Notice

that in the above expression we have used $p_{\text{vac}} = -\rho_{\text{vac}}$ for the EoS of the vacuum fluid, as indicated in the introduction. Even though this condition may be violated slightly by quantum effects within a formal treatment of the subject in QFT [31], we shall nonetheless stick for now to the traditional EoS of the vacuum. We shall come back to this point later on. Upon expanding $\nabla^{\mu} T^{\text{tot}}_{\mu\nu} = 0$, it amounts to the local covariant conservation law in an FLRW universe:

$$\frac{d}{dt}[G(\rho_t + \rho_{\text{vac}})] + 3GH(\rho_t + p_t) = 0, \tag{5}$$

where, in general, not only $\rho_{\text{vac}}$ but also $G$ may be functions of the cosmic time. This will depend on the particular implementation assumed for the matter sector. If we assume that there is an interaction of the VED with matter, then $G$ can stay fixed at the usual value $G_N$ (the local gravity value), whereas if matter is locally conserved, then $G$ must vary accordingly in order to preserve the covariant conservation law (5).

In order to ease the comparison with previous results, we shall adhere to the approach of [85] and assume $\tilde{\nu} = \nu/2$. In this way, the RVM model is left with one single parameter and at the same time adopts the suggestive form

$$\rho_{\text{vac}}(H) = \frac{3}{8\pi G_N}\left(c_0 + \frac{\nu}{12}\mathcal{R}\right) \equiv \rho_{\text{vac}}(\mathcal{R}), \tag{6}$$

in which $\mathcal{R} = 12H^2 + 6\dot{H}$ is the curvature scalar. That particular implementation is called, for obvious reasons, the RRVM, since it is a version of the RVM which involves the scalar of curvature [85]. One additional advantage is that it is automatically well-behaved in the radiation-dominated epoch, since in it $\mathcal{R}/H^2 \ll 1$ and the standard BBN is not perturbed at all by the presence of vacuum energy. In the general case (1), such a condition can also be fulfilled by assuming sufficiently small (absolute) values of $\nu$ and $\tilde{\nu}$ [90].

Finally, despite the general structure of the running VED being of the form (1), for convenience we define two types of RRVM scenarios. In the type-I scenario, the vacuum is in interaction with matter, whereas in the type-II scenario, matter is conserved at the expense of an exchange between the vacuum and a slowly evolving gravitational coupling $G(H)$. The combined cosmological 'running' of these quantities ensures the accomplishment of the Bianchi identity (and the associated local conservation law). In the following sections we study these two cases separately.

## 3. Type I: Running Vacuum Interacting with Dark Matter

In this section, we consider the type-I RRVM scenario, in which the vacuum can be running at the expense of exchanging energy with matter. We will assume that only cold dark matter (CDM) is involved in such an exchange (therefore no baryons, neutrinos or photons are transferred to or from the vacuum). Whether it is the vacuum that generates new CDM or the CDM that disappears into the vacuum depends on the sign of the parameter $\nu$ in Equation (6). For $\nu > 0$, the vacuum decays into tiny amounts of CDM, whilst for $\nu < 0$ some dark matter disappears into the vacuum. We do not presume which of these situations holds, we will fit the value (and sign) of $\nu$ to the cosmological data. This requires solving the background and linear perturbation equations of the type-I running vacuum model, which we demonstrate in Sections 3.1 and 3.2.

### 3.1. Background Equations

The VED expression (6) can be cast more explicitly as follows,

$$\rho_{\text{vac}} = \frac{3}{8\pi G_N}\left[c_0 + \nu\left(H^2 + \frac{1}{2}\dot{H}\right)\right]. \tag{7}$$

The above energy component now becomes a part of the Friedmann and the pressure equations written in terms of the energy densities and the pressures for the different species, which read

$$3H^2 = 8\pi G_N(\rho_t + \rho_{\text{vac}}) = 8\pi G_N(\rho_m + \rho_{\text{ncdm}} + \rho_\gamma + \rho_{\text{vac}}), \tag{8}$$

$$3H^2 + 2\dot{H} = -8\pi G_N(p_t + p_{\text{vac}}) = -8\pi G_N(p_{\text{ncdm}} + p_\gamma + p_{\text{vac}}). \tag{9}$$

The following comment is in order. As is well known, there is a transfer of energy from the relativistic neutrinos to the nonrelativistic ones throughout the whole of cosmic history. It is difficult to make a perfect separation of the relativistic and nonrelativistic phases and, strictly speaking, this splitting can be a little bit inaccurate at those epochs of the expansion history for which a neutrino species is in an intermediate step, between the ultra-relativistic and nonrelativistic regimes, since in this case one cannot classify such neutrino species in any of these two categories. Nevertheless, it is useful to obtain approximate formulas for the two components, as we shall see in a moment. We proceed as in the Einstein–Boltzmann solver CLASS[1] [102,103], where we have implemented our model. CLASS solves the Einstein and Boltzmann differential equations at any value of the scale factor and, in particular, provides the functions $\rho_{\text{ncdm}}(a)$ and $p_{\text{ncdm}}(a)$. CLASS then performs a rather artificial splitting of these quantities, as if they came from the sum of an ultra-relativistic fluid (denoted with a subscript $\nu$) and a nonrelativistic one (denoted with a subscript $h$),

$$\rho_h = \rho_{\text{ncdm}} - 3p_{\text{ncdm}} \quad ; \quad p_h = 0; \tag{10}$$

$$\rho_\nu = 3p_{\text{ncdm}} \quad ; \quad p_\nu = p_{\text{ncdm}}. \tag{11}$$

At this point, we can rewrite the combination $H^2 + (1/2)\dot{H}$ appearing in (7) in terms of the energy densities and pressures using (8) and (9),

$$H^2 + \frac{1}{2}\dot{H} = \frac{2\pi G_N}{3}(\rho_m + 4\rho_{\text{vac}} + \rho_{\text{ncdm}} - 3p_{\text{ncdm}}). \tag{12}$$

In this expression, we can appreciate that the nonrelativistic contribution from massive neutrinos, namely $\rho_h = \rho_{\text{ncdm}} - 3p_{\text{ncdm}}$, is present. This is a problem if we want to solve the background equations, since it carries a complicated (non-analytic) dependence on the scale factor. In order to solve this problem, we can consider a reasonable approximation, which is the following:

$$r \equiv \frac{\rho_h}{\rho_m} = \frac{\rho_h}{\rho_{\text{cdm}} + \rho_b} \simeq 0. \tag{13}$$

We have checked explicitly the validity of this approximation with CLASS, computing the ratio $r = \rho_h/\rho_m$ for the whole of cosmic history. We have found that $r$ varies smoothly from $10^{-7}$ at redshift $z = 10^{14}$ to $10^{-3}$ at $z = 0$, considering a massive neutrino with mass $\sim \mathcal{O}(0.1)$ eV. In addition, $r$ is multiplied by $\nu$ in Equation (7), so the resulting quantity is of the order $\mathcal{O}(10^{-5})$ at most. Therefore, we deem it natural and licit to drop this term to make things easier without any significant loss in accuracy in our calculation.

Under this very good approximation, we can express the vacuum energy density (7) as follows,

$$\rho_{\text{vac}}(a) = \rho_{\text{vac}}^0 + \frac{\nu}{4(1-\nu)}(\rho_m(a) - \rho_m^0), \tag{14}$$

with $\rho_{\text{vac}}(a = 1) = \rho_{\text{vac}}^0$ and $\rho_m(a = 1) = \rho_m^0$. We still need to find $\rho_m(a)$ though. The starting point is Equation (5) which yields the interaction law between vacuum and matter in the general case. Now, since we assume that $G$ is strictly constant for the type-I models, Equation (5) can be reduced to

$$\dot{\rho}_m + 3H\rho_m = -\dot{\rho}_{\text{vac}}, \tag{15}$$

where we are neglecting the pressure of the matter components. Notice that the previous equation is entirely equivalent to the interaction law between CDM and the vacuum,

$$\dot{\rho}_{\mathrm{cdm}} + 3H\rho_{\mathrm{cdm}} = -\dot{\rho}_{\mathrm{vac}}, \tag{16}$$

owing to the fact that we are assuming that baryons do not interact at all with the vacuum, which entails the relation $\dot{\rho}_b + 3H\rho_b = 0$. In this way we have obtained the conservation equation of matter (baryons+CDM) for type-I models.

Combining the above equations, we arrive at the final result

$$\dot{\rho}_m + 3H\xi\rho_m = 0, \tag{17}$$

where we have defined the dimensionless parameter

$$\xi \equiv \frac{1-\nu}{1-\frac{3}{4}\nu}. \tag{18}$$

It is then straightforward to find out the expressions for the various energy densities:

$$\rho_m(a) = \rho_m^0 a^{-3\xi}, \tag{19}$$

$$\rho_{\mathrm{cdm}}(a) = \rho_m^0 a^{-3\xi} - \rho_b^0 a^{-3}, \tag{20}$$

$$\rho_{\mathrm{vac}}(a) = \rho_{\mathrm{vac}}^0 + \left(\frac{1}{\xi} - 1\right)\rho_m^0\left(a^{-3\xi} - 1\right). \tag{21}$$

In the limit $\xi \to 1$ ($\nu \to 0$) we recover the expected forms of these equations in the $\Lambda$CDM. It is also possible to encode the deviations with respect to the standard cosmological model in terms of an effective parameter $\nu_{\mathrm{eff}}$, defined as

$$\xi = \frac{1-\nu}{1-\frac{3}{4}\nu} \simeq 1 - \frac{\nu}{4} + \mathcal{O}\left(\nu^2\right) \equiv 1 - \nu_{\mathrm{eff}} + \mathcal{O}\left(\nu_{\mathrm{eff}}^2\right). \tag{22}$$

We will report all our fitting results in terms of parameter $\nu_{\mathrm{eff}}$. As with $\nu$, positive values of $\nu_{\mathrm{eff}}$ lead to a decay of the vacuum into CDM, whereas negative values source an energy transfer from CDM to the vacuum.

### 3.2. Perturbation Equations

We have implemented the perturbation equations in CLASS, using the synchronous gauge. Denoting by $\tau$ the conformal time, the perturbed (flat three-dimensional) FLRW metric in the conformal frame reads [104],

$$ds^2 = a^2(\tau)[-d\tau^2 + (\delta_{ij} + h_{ij})dx^i dx^j], \tag{23}$$

with

$$h_{ij}(\tau, \vec{x}) = \int d^3k e^{-i\vec{k}\cdot\vec{x}}\left[\hat{k}_i\hat{k}_j h(\tau, \vec{k}) + \left(\hat{k}_i\hat{k}_j - \frac{\delta_{ij}}{3}\right)6\eta(\tau, \vec{k})\right], \tag{24}$$

and $\hat{k}_i = k_i/k$. The above formula represents the perturbation expressed as a Fourier integral on the two fields in $k$-space, $h(\tau, \vec{k})$ and $\eta(\tau, \vec{k})$, which parameterize the non-traceless and traceless parts, respectively. The nonvanishing trace is the $h$ function. The perturbed Einstein equations in Fourier space adopt the same form as in the $\Lambda$CDM. They read as follows:

$$\mathcal{H}h' - 2\eta k^2 = 8\pi G_N a^2 \sum_l \delta\rho_l, \tag{25}$$

$$\eta' k^2 = 4\pi G_N a^2 \sum_l (\bar{\rho}_l + \bar{p}_l)\theta_l, \tag{26}$$

$$h'' + 2\mathcal{H}h' - 2\eta k^2 = -24\pi G_N a^2 \sum_l \delta p_l, \tag{27}$$

$$h'' + 6\eta'' + 2\mathcal{H}(h' + 6\eta') - 2k^2\eta = -24\pi G_N a^2(\bar{\rho} + \bar{p})\sigma, \tag{28}$$

where $\mathcal{H} \equiv aH$, the sums run over the different matter components, the primes denote derivatives with respect to the conformal time, and

$$(\bar{\rho} + \bar{p})\sigma \equiv -\left(\hat{k}_i\hat{k}_j - \frac{\delta_{ij}}{3}\right)\left(T_j^i - \frac{\delta_j^i}{3}T_k^k\right) \tag{29}$$

carries the information of the anisotropic stress, with $T_{\mu\nu}$ the total energy–momentum tensor. The bars in these equations indicate background quantities and $\theta_l$ is the divergence of the perturbed velocity of the fluid $l$. Equations (25) and (26) are obtained from the 00 and $0i$ components of Einstein's equations, respectively, whereas Equations (27) and (28) are the trace and the part proportional to $\hat{k}_i\hat{k}_j$ of the $ij$ component.

All the perturbed conservation equations are also the same as in the standard model, except those that relate CDM and the vacuum, which take the following form,

$$\theta'_{\text{cdm}} + \mathcal{H}\theta_{\text{cdm}} = \frac{\bar{\rho}'_{\text{vac}}}{\bar{\rho}_{\text{cdm}}}\theta_{\text{cdm}} - k^2\frac{\delta\rho_{\text{vac}}}{\bar{\rho}_{\text{cdm}}}, \tag{30}$$

$$\delta'_{\text{cdm}} - \frac{\bar{\rho}'_{\text{vac}}}{\bar{\rho}_{\text{cdm}}}\delta_{\text{cdm}} + \frac{\delta\rho'_{\text{vac}}}{\bar{\rho}_{\text{cdm}}} + \theta_{\text{cdm}} + \frac{h'}{2} = 0, \tag{31}$$

with $\delta_{\text{cdm}} = \delta\rho_{\text{cdm}}/\bar{\rho}_{\text{cdm}}$ the CDM density contrast and $\theta_{\text{cdm}}$ the divergence of the perturbed CDM 3-velocity. We consider a vacuum–geodesic CDM interaction such that there is no net momentum transfer between the vacuum and cold dark matter [41,105,106]. Thus, we can fix the gauge by setting $\theta_{\text{cdm}} = 0$, as in the $\Lambda$CDM. This automatically sets $\delta\rho_{\text{vac}} = 0$. In this setup, Equation (31) simplifies to

$$\delta'_{\text{cdm}} + \frac{h'}{2} - \frac{\bar{\rho}'_{\text{vac}}}{\bar{\rho}_{\text{cdm}}}\delta_{\text{cdm}} = 0. \tag{32}$$

This is actually the only perturbation equation that must be modified in CLASS in order to accommodate the dynamical character of the VED, i.e., the fact that $\bar{\rho}'_{\text{vac}} \neq 0$. In this work, we consider adiabatic perturbations for the various matter and radiation species.

### 3.3. Type I with Threshold

Once we have obtained the background and the perturbation equations we are in a position to study the cosmological evolution of the RVM in different situations. The conventional option would be to assume that the above equations are valid throughout the entire cosmic history (subsequent, of course, to the inflationary period, which will not be dealt with here at all). The phenomenological analyses of [88–93], for example, were based on that standard assumption. However, we may also entertain the intriguing possibility that the interaction between the vacuum and dark matter is only relatively recent. In that case, we could have a scenario where the dynamics of the vacuum starts approximately at the time when it becomes dominant over matter (i.e., at about the outset of what is usually referred to as the DE epoch). Such a situation should be characterized by the presence of a 'threshold point' for the vacuum dynamics at some redshift value $z_*$, where the transition occurs. The idea is to study the response of our fit to the overall cosmological data when we switch off the interaction between the VED and the CDM for most of cosmic history, except when we approach the usual epoch of vacuum dominance. Since the conventional DE epoch in the late universe is usually assumed to commence at around a redshift value of $z_* \simeq 1$, we will assume that its evolution also begins at around that point (see below).

We will implement the simplest version of such a threshold scenario through a Heaviside $\Theta$-function, and for definiteness it will be restricted to type-I models only. Thus, let

$a_*$ be the value of the scale factor where the activation of the vacuum dynamics occurs ($z_* = a_*^{-1} - 1$ being the corresponding redshift value). Before reaching that point (that is, at earlier epochs $a < a_*$) the vacuum is rigid, whereas after that point (hence nearer to our present time) the vacuum evolves with the expansion following the type-I running vacuum behavior; see Equation (21). We have checked that the optimal value for this parameter is $a_* \simeq 0.5$, which indeed corresponds to $z_* \simeq 1$.[2]

It should be noted that while the VED function $\rho_{vac}(a)$ will remain continuous in our implementation of the step function procedure, its time derivative $\dot{\rho}_{vac}$ does not, and in fact it is modified through a Heaviside function factor $\Theta(a - a_*)$. Accordingly, the derivative of the CDM energy density must also change in a discontinuous way. In contradistinction, all the energy densities are continuous at $a = a_*$. Consequently, we find that in order to fulfill these requirements we must implement the analytical expressions for the various density functions in the following way:

$\underline{a < a_* \ (z > z_*)}$

$$\rho_{cdm}(a) = \rho_{cdm}(a_*)\left(\frac{a}{a_*}\right)^{-3} \tag{33}$$

$$\rho_{vac}^* = \rho_{vac}^0 + \left(\frac{1}{\xi} - 1\right)\rho_m^0\left(a_*^{-3\xi} - 1\right) = \text{const.} \tag{34}$$

$\underline{a > a_* \ (z < z_*)}$

$$\rho_{cdm}(a) = \rho_m^0 a^{-3\xi} - \rho_b^0 a^{-3} \tag{35}$$

$$\rho_{vac}(a) = \rho_{vac}^0 + \left(\frac{1}{\xi} - 1\right)\rho_m^0\left(a^{-3\xi} - 1\right). \tag{36}$$

In the above expression, we have defined

$$\rho_{cdm}(a_*) = \rho_m^0 a_*^{-3\xi} - \rho_b^0 a_*^{-3} = \text{const.} \tag{37}$$

It goes without saying that the same modifications have to be applied in the perturbation sector. We denote this version of the type-I RRVM with threshold as type-I RRVM$_{thr.}$.

We do not wish to speculate here on the possible origin of the threshold postulated above, it could be a manifestation of a late-time interaction in the dark sector. However, we mention that a fundamental microscopical explanation might come from the RVM framework emerging from QFT in curved spacetime [31], which indicates that the EoS of the quantum vacuum stays in the characteristic DE range ($0 < w_{vac} < -1/3$) only below a redshift value (threshold) in our recent past $z_* \simeq 1$. For $z > z_*$, instead, one has $w_{vac} > -1/3$ and the vacuum no longer behaves as DE. This is of course impossible for the classical vacuum, for which $w_{vac} = -1$ all the time. Additional studies will obviously be necessary to gauge the impact of such EoS behavior on the global fits to the cosmological data and its potential relation with the type-I scenario with threshold that we have defined above. Finally, we mention that the behavior of the type-I RRVM with threshold should be essentially the same as that of the 'canonical RVM' with threshold, namely, the original RVM form with the dynamical component $\sim H^2$; see [17,19] and references therein. The latter corresponds to (1) with only the single parameter $\nu$ (with $\tilde{\nu} = 0$). At low $z$, the two models are expected to be indistinguishable from the phenomenological point of view.

## 4. Type II: Running Vacuum with Running $G$

For type-II RRVM we have an entirely different sort of scenario, in which matter is strictly conserved, in particular dark matter, and hence no interaction of any sort is per-

mitted between matter and vacuum. However, to switch off the energy exchange between matter and vacuum in a fully consistent way with the Bianchi identity, we must allow for the running of the gravitational coupling with the expansion, $G = G(H)$. Therefore, for type-II models we have both the running of the vacuum energy density $\rho_{\rm vac}$ and the running of $G$. Let us briefly see how this comes about. If matter is conserved, we have $\dot{\rho}_t + 3H(\rho_t + p_t) = 0$, where as in the previous sections the subscript $t$ refers to the sum of all the species in the universe excluding the vacuum. Whereupon the general Bianchi identity (5) can be reduced to[3]

$$\dot{G}(\rho_t + \rho_{\rm vac}) + G\dot{\rho}_{\rm vac} = 0. \tag{38}$$

Since the running of $\rho_{\rm vac}$ is still fixed by (7), the previous equation is essential to determine the running of $G$. Being $\dot{\rho}_{\rm vac} \propto \nu$, the sign of $\nu$ determines the sign of $\dot{G}$, i.e., if $\nu > 0$ ($\nu < 0$) $G$ increases (decreases) with the cosmic expansion. Of course, we have to make sure that $G$ evolves in a very mild way, which in fact turns out to be the case as we shall verify explicitly.

In what follows, we bring forth the relevant background and linear perturbation equations for the type-II RRVM in Sections 4.1 and 4.2, respectively. As will become clear, the solution of this model type is more complicated.

### 4.1. Background Equations

We come back to the following form of the vacuum energy density,

$$\rho_{\rm vac}(\mathcal{R}) = C_0 + \frac{\nu}{32\pi G_N}\mathcal{R}. \tag{39}$$

which is of course equivalent to our original RRVM expression (6), with $C_0 \equiv 3c_0/(8\pi G_N)$ and $\mathcal{R} = 12H^2 + 6\dot{H}$. The Friedmann and pressure equations read, respectively,

$$3H^2 = 8\pi G\left[\rho_t + C_0 + \frac{3\nu}{16\pi G_N}(2H^2 + \dot{H})\right], \tag{40}$$

$$-(3H^2 + 2\dot{H}) = 8\pi G\left[p_t - C_0 - \frac{3\nu}{16\pi G_N}(2H^2 + \dot{H})\right], \tag{41}$$

where $G_N$ is Newton's gravitational constant, whereas $G$ stands for the running gravitational coupling. For type-II models $G$ evolves with the expansion, and hence it is generally different from $G_N$. At the same time, we remind the reader that for type-II models the background energy densities and pressures of the matter species evolve as a function of the scale factor exactly as in the $\Lambda$CDM, since now there is no energy exchange between them and the vacuum.

A practical consideration is now in order, which will make clear immediately why solving the type-II models is more difficult. Recall that for the numerical analysis we are using the CLASS system solver [102,103]. Now, the point is that for the standard cosmological model, CLASS computes $H$ and $\dot{H}$ after computing the energy densities of the various components that fill the universe. In the model under consideration though, we cannot proceed in the same way, because we first need to compute $G$. Before explaining how it is still possible to solve the system in the CLASS platform, it is useful to rewrite the above equations in terms of the auxiliary variable[4] $\varphi \equiv G_N/G$. One expects $\varphi \simeq 1$ at present and one may even impose this condition (see, however, below). In terms of $\varphi$, the set of relevant equations read

$$3H^2 = \frac{8\pi G_N}{\varphi}\left[\rho_t + C_0 + \frac{3\nu}{16\pi G_N}(2H^2 + \dot{H})\right], \tag{42}$$

$$-(3H^2 + 2\dot{H}) = \frac{8\pi G_N}{\varphi}\left[p_t - C_0 - \frac{3\nu}{16\pi G_N}(2H^2 + \dot{H})\right], \tag{43}$$

$$\frac{\dot{\varphi}}{\varphi} = \frac{\dot{\rho}_{\text{vac}}}{\rho_t + \rho_{\text{vac}}}, \tag{44}$$

where the last one, Equation (44), is of course nothing but a rephrasing of Equation (38). This equation is rather complicated since it is obvious from (7) that it involves not only $H$ and $\dot{H}$, but also $\ddot{H}$. In order to compute the latter, it is possible, of course, to differentiate the pressure equation and use it together with (42) and (43). Unfortunately, in doing so one obtains $\ddot{H}$ as a function of the derivative of the neutrinos' energy density and pressure, which should then be computed numerically. This approach looks too complicated, and therefore we opt for the following alternative and simpler method. We first obtain a differential equation for $H$. Dividing out Equations (42) and (43) we can lose $\varphi$, and after some rearrangement we are led to the following differential equation:

$$0 = \frac{3\nu}{8\pi G_N}\dot{H}^2 + \dot{H}\left(2C_0 + 2\rho_t + \frac{3\nu H^2}{4\pi G_N}\right) + 3H^2(\rho_t + p_t). \tag{45}$$

The previous equation can be restated in the more convenient form

$$\dot{H} = \frac{4\pi G_N}{3\nu}\left(-B + \sqrt{B^2 - \nu\frac{9H^2}{2\pi G_N}(p_t + \rho_t)}\right), \tag{46}$$

where we have defined the function

$$B \equiv 2C_0 + 2\rho_t + \frac{3\nu H^2}{32 G_N}. \tag{47}$$

Equation (46) can be solved much more easily than (44), although we still need to employ a numerical procedure. We have all the necessary ingredients. As an initial condition (at $z_{\text{ini}} \sim 10^{14}$) we can use

$$H_{\text{ini}}^2 = \frac{8\pi G_N}{3\varphi_{\text{ini}}}\rho_r(z_{\text{ini}}), \tag{48}$$

because the radiation energy density clearly dominates over the nonrelativistic matter and the vacuum. We can tell `CLASS` to apply the finite difference method to solve the system step by step. In each of these steps `CLASS` computes $H_{n+1}$ from $H_n$ and $\dot{H}_n$ (46). For the latter it takes the various energy densities and pressures. Then, we can employ the Friedmann equation to compute $\varphi_{n+1}$,

$$\varphi = \frac{8\pi G_N}{3H^2}\left[\rho_t + C_0 + \frac{3\nu}{16\pi G_N}(2H^2 + \dot{H})\right], \tag{49}$$

and iterate the process till we have the complete expansion history to the necessary degree of accuracy.

Next, we show the evolution of $\varphi$ during the radiation-dominated (RDE) epoch. First, let us write $\rho_{\text{vac}}$ in terms of the energy densities, pressures, and $\varphi$. Notice that using (42) and (43) we obtain:

$$H^2 = \frac{8\pi G_N}{3(\varphi - \nu)}\left[\rho_t + C_0 - \frac{3}{4}\frac{\nu}{\varphi}(\rho_t + p_t)\right], \tag{50}$$

$$\dot{H} = -\frac{4\pi G_N}{\varphi}(\rho_t + p_t). \tag{51}$$

Introducing these expressions into (39) we obtain:

$$\rho_{\text{vac}}(a) = \frac{\varphi(a)C_0 + \frac{\nu}{4}[\rho_t(a) - 3p_t(a)]}{\varphi(a) - \nu}. \tag{52}$$

Even though we have solved the type-II model in an exact way using the aforementioned numerical strategy, the following approximate analytical considerations may help to better understand the behavior of the solution. In the RDE, we have

$$\rho_{\text{vac}}(a) = \frac{\nu \rho_m^0}{4\varphi(a)} a^{-3} + \mathcal{O}(\nu^2) \tag{53}$$

since only the nonrelativistic component $\rho_m = \rho_m^0 a^{-3}$ contributes in the term proportional to $\nu$ in the numerator of (52) after the pressure and the radiation densities cancel in the difference $\rho_r(a) - 3p_r(a)$. Finally, using (53) in (44) we can easily integrate the resulting equation, since in the denominator we have $\rho_{\text{vac}}(a) \ll \rho_t(a) \simeq \rho_r^0 a^{-4}$ in the radiation epoch. Integrating from the initial scale factor value $a_{\text{ini}} = 10^{-14}$ (see above) up to an arbitrary value, we find the evolution of the gravitational coupling within this analytical approximation:

$$\varphi(a) = \varphi_{\text{ini}} - \frac{3\nu_{\text{eff}}}{a_{\text{eq}}}(a - a_{\text{ini}}), \tag{54}$$

with $a_{\text{eq}} = \Omega_{r,*}^0 / \Omega_m^0$ and $\Omega_{r,*}^0$ being the radiation density parameter computed assuming that the neutrinos are all massless. Similarly to the type-I RRVM (cf. Section 3.1), we have defined

$$\nu_{\text{eff}} \equiv \frac{\nu}{4}, \tag{55}$$

as in [85]. We actually report the fitting value of this parameter in our tables and contour plots; see the discussion in Section 6. The term $(a - a_{\text{ini}})/a_{\text{eq}}$ in (54) is much smaller than 1, since (54) is valid for $a \ll a_{\text{eq}}$. Hence, the total variation in $\varphi$ during the RDE is small and of the order $\nu_{\text{eff}}$, $\Delta\varphi \approx -3\nu_{\text{eff}} \sim \mathcal{O}(\nu)$, despite being linear with the scale factor. It is easy to see that the relation (53) still holds in the MDE, but the denominator of (44) is now dominated by the term $\rho_t(a) \simeq \rho_m^0 a^{-3}$. Integration now gives

$$\varphi(a) = C - 3\nu_{\text{eff}} \ln a, \tag{56}$$

where $C$ is a constant to be fixed by some initial condition in the MDE. It is easy to check that the total variation in $\varphi$ in the MDE will be of the order of $\sim \mathcal{O}(10)\nu$, i.e., ten times larger than in the RDE. In the vacuum-dominated epoch, the right-hand side of Equation (44) goes to zero and $\varphi \to \text{const}$, so $G$ remains constant as well.

Traditional limits on the relative time variation in $G$ can be found, e.g., in the review [114]. More recent determinations, e.g., those based on measurements on the double pulsar PSR J0737–3039A/B, yield rather tight bounds [115]:

$$\frac{\dot{G}}{G} = (-0.8 \pm 1.4) \times 10^{-13} \frac{1}{\mathcal{F}_{AB}} \text{yr}^{-1}, \tag{57}$$

where $\mathcal{F}_{AB} \simeq 1$ for weakly self-gravitating bodies. A previous limit from a binary pulsar (PSR J1713+0747) provided [116]

$$\frac{\dot{G}}{G} = (-0.1 \pm 0.9) \times 10^{-12} \text{yr}^{-1}, \tag{58}$$

which is a bit weaker but essentially in the same ballpark. On the other hand, the best current limit on the relative variation in $G$ obtained in the solar system is [117]

$$\frac{\dot{G}}{G} < 4 \times 10^{-14} \text{yr}^{-1}. \tag{59}$$

Assuming that (57)–(59) can be used to constrain the cosmological evolution of $G$, let us tentatively use these limits (in order of magnitude) in combination with Equation (56) to constrain the parameter $\nu_{\text{eff}}$. The last equation implies $\dot{\varphi} \simeq -3\nu_{\text{eff}}H_0$ for $H \simeq H_0 \simeq 1.023h \times 10^{-10}$ yr$^{-1}$ around our time ($h \simeq 0.7$) The previous relation is equivalent to $\dot{G}/G \simeq 3\nu_{\text{eff}}H_0$ to order $\nu_{\text{eff}}$. It is then easy to check that in order to fulfill the above limits we must require $\nu_{\text{eff}} \lesssim (2-5) \times 10^{-4}$, a condition which is satisfied by our fitting values of $\nu_{\text{eff}}$ for the type-II models, even in the most restrictive case (see the fitting tables of Section 6)[5].

From the above considerations it is clear that $\varphi = G_N/G$ changes very slowly (logarithmically) with the expansion, and proportionally to the small parameter $\nu$. The predicted variations in $G$ within the type-II running vacuum models lie, in fact, within the most restrictive experimental limits existing in the literature. This demonstrates our contention, mentioned previously, that the running of $G$ within these models is consistent with the observations.

Finally, we mention that with the purpose of giving more freedom to the model, in this work we will not impose the condition $G(a = 1) = G_N$ or, equivalently, $\varphi(a = 1) = 1$, as we did in previous studies such as [85] and also in [118–120], within the context of the Brans–Dicke model with a cosmological constant. We naturally expect $\varphi(a = 1) \simeq 1$, of course. The obtained fitting values for $\varphi$ at present are indicated in our tables as $\varphi(0) \equiv \varphi(z = 0)$. Let us finally note that cosmologies with variable $G$ may have to rely on an efficient screening mechanism that allows the recovery of $G_N$ at the solar system. We will not focus on this issue, which has been addressed in several places in the literature (see, e.g., [4,121,122] and references therein, and more recently in [123]), as it deserves a more devoted study which is certainly beyond the scope of this work. Let us, however, note that this situation affects mainly the Brans–Dicke-type models [113], where the effective gravitational coupling is tied to a dynamical scalar field that could mediate long-range interactions as a true degree of freedom of the underlying gravitational framework. As indicated before, this is not our case. These situations may have an impact only in the non-linear scales, which are anyway not strongly affected by the cosmological observables employed in this study. We close this section by noting that, while in this paper we have assumed that $G$ may change with the cosmic time in type-II models, it could also evolve with a distance scale in a galactic domain, $G = G(r)$ ($0 < r < L$). Such an extension of the running of the gravitational coupling was considered in [124] and could be helpful to connect the running of $G$ between the galactic/astrophysical and cosmological domains.

**Table 1.** Published values of BAO data; see the quoted references for details and for the corresponding covariance matrices. The fiducial values of the comoving sound horizon appearing in the table are $r_{d,\text{fid}} = 147.5$ Mpc for [125] and $r_{d,\text{fid}} = 148.6$ Mpc for [126].

| Survey | $z$ | Observable | Measurement | References |
|:---:|:---:|:---:|:---:|:---:|
| 6dFGS+SDSS MGS | 0.122 | $D_V(r_d/r_{d,\text{fid}})$ [Mpc] | $539 \pm 17$ [Mpc] | [125] |
| DR12 BOSS | 0.32 | $Hr_d/(10^3\text{km/s})$ | $11.549 \pm 0.385$ | [127] |
| | | $D_A/r_d$ | $6.5986 \pm 0.1337$ | |
| | 0.57 | $Hr_d/(10^3\text{km/s})$ | $14.021 \pm 0.225$ | |
| | | $D_A/r_d$ | $9.389 \pm 0.1030$ | |

**Table 1.** *Cont.*

| Survey | $z$ | Observable | Measurement | References |
|---|---|---|---|---|
| WiggleZ | 0.44 | $D_V(r_d/r_{d,\text{fid}})$ [Mpc] | $1716.4 \pm 83.1$ [Mpc] | [126] |
| | 0.60 | $D_V(r_d/r_{d,\text{fid}})$ [Mpc] | $2220.8 \pm 100.6$ [Mpc] | |
| | 0.73 | $D_V(r_d/r_{d,\text{fid}})$ [Mpc] | $2516.1 \pm 86.1$ [Mpc] | |
| DESY3 | 0.835 | $D_M/r_d$ | $18.92 \pm 0.51$ | [128] |
| eBOSS Quasar | 1.48 | $D_M/r_d$ | $30.21 \pm 0.79$ | [129] |
| | | $D_H/r_d$ | $13.23 \pm 0.47$ | |
| Ly$\alpha$-Forests | 2.334 | $D_M/r_d$ | $37.5^{+1.2}_{-1.1}$ | [130] |
| | | $D_H/r_d$ | $8.99^{+0.20}_{-0.19}$ | |

**Table 2.** Values of $H(z)$ from cosmic chronometers and their $1\sigma$ uncertainties, which include the contribution of statistical and systematic effects [131]. They are expressed in km/s/Mpc. We have considered the correlations between the data points marked with a *, as discussed in [131]. In some of the quoted references, the authors provide measurements obtained with two different stellar population synthesis (SPS) models. In these cases, we have employed the mean of the two central values and statistical errors. The systematic uncertainty already accounts for the choice of SPS model.

| $z$ | $H(z)$[km/s/Mpc] | References |
|---|---|---|
| 0.07 | $69.0 \pm 19.6$ | [132] |
| 0.09 | $69.0 \pm 12.0$ | [133] |
| 0.12 | $68.6 \pm 26.2$ | [132] |
| 0.17 | $83.0 \pm 8.0$ | [134] |
| 0.1791 * | $77.72 \pm 6.01$ | [135] |
| 0.1993 * | $77.79 \pm 6.83$ | [135] |
| 0.2 | $72.9 \pm 29.6$ | [132] |
| 0.27 | $77.0 \pm 14.0$ | [134] |
| 0.28 | $88.8 \pm 36.6$ | [132] |
| 0.3519 * | $85.45 \pm 15.75$ | [135] |
| 0.3802 * | $86.17 \pm 14.61$ | [136] |
| 0.4 | $95.0 \pm 17.0$ | [134] |
| 0.4004 * | $79.90 \pm 11.38$ | [136] |
| 0.4247 * | $90.39 \pm 12.76$ | [136] |
| 0.4497 * | $96.24 \pm 14.38$ | [136] |
| 0.47 | $89.0 \pm 49.6$ | [137] |
| 0.4783 * | $83.74 \pm 10.18$ | [136] |
| 0.48 | $97.0 \pm 62.0$ | [138] |
| 0.5929 * | $106.80 \pm 15.06$ | [135] |
| 0.6797 * | $94.875 \pm 10.600$ | [135] |
| 0.75 | $89.0 \pm 49.6$ | [139] |
| 0.7812 * | $96.27 \pm 12.72$ | [135] |
| 0.8754 * | $124.70 \pm 17.13$ | [135] |

**Table 2.** *Cont.*

| $z$ | $H(z)$[km/s/Mpc] | References |
|---|---|---|
| 0.88 | $90.0 \pm 40.0$ | [138] |
| 0.9 | $117.0 \pm 23.0$ | [134] |
| 1.037 * | $133.35 \pm 18.12$ | [135] |
| 1.3 | $168.0 \pm 17.0$ | [134] |
| 1.363 * | $163.95 \pm 34.61$ | [140] |
| 1.43 | $177.0 \pm 18.0$ | [134] |
| 1.53 | $140.0 \pm 14.0$ | [134] |
| 1.75 | $202.0 \pm 40.0$ | [134] |
| 1.965 * | $191.10 \pm 51.91$ | [140] |

**Table 3.** Published values of $f(z)\sigma_8(z)$; see the quoted references and text in Section 5.

| Survey | $z$ | $f(z)\sigma_8(z)$ | References |
|---|---|---|---|
| ALFALFA | 0.013 | $0.46 \pm 0.06$ | [141] |
| 6dFGS+SDSS | 0.035 | $0.338 \pm 0.027$ | [142] |
| GAMA | 0.18 | $0.29 \pm 0.10$ | [143] |
| | 0.38 | $0.44 \pm 0.06$ | [144] |
| WiggleZ | 0.22 | $0.42 \pm 0.07$ | [145] |
| | 0.41 | $0.45 \pm 0.04$ | |
| | 0.60 | $0.43 \pm 0.04$ | |
| | 0.78 | $0.38 \pm 0.04$ | |
| DR12 BOSS | 0.32 | $0.427 \pm 0.056$ | [127] |
| | 0.57 | $0.426 \pm 0.029$ | |
| VIPERS | 0.60 | $0.49 \pm 0.12$ | [146] |
| | 0.86 | $0.46 \pm 0.09$ | |
| VVDS | 0.77 | $0.49 \pm 0.18$ | [147,148] |
| FastSound | 1.36 | $0.482 \pm 0.116$ | [149] |
| eBOSS Quasar | 1.48 | $0.462 \pm 0.045$ | [129] |

### 4.2. Perturbation Equations

The actual implementation of the linear perturbation equations for type-II models in the `CLASS` computing platform is more difficult than the background part (which was already nontrivial), and certainly much more involved than the one carried out for the type-I variant (viz. the one with vacuum–CDM interaction). We use again the synchronous gauge, but in this case the gauge unfortunately does not fix $\delta\rho_{\mathrm{vac}} = 0$, in contrast to the situation in Section 3.2, so we have to keep the contribution of the vacuum perturbation in our equations.

The 00, 0*i*, and *ii* components of the Einstein equations in Fourier space read, respectively,

$$\mathcal{H}h' - 2\eta k^2 = 8\pi G_N a^2 \sum_l \left( \frac{\delta\rho_l}{\bar{\varphi}} - \bar{\rho}_l \frac{\delta\varphi}{\bar{\varphi}^2} \right), \tag{60}$$

$$\eta' k^2 = \frac{4\pi G_N}{\bar{\varphi}} a^2 \sum_l (\bar{\rho}_l + \bar{p}_l)\theta_l, \tag{61}$$

$$h'' + 2\mathcal{H}h' - 2\eta k^2 = \frac{24\pi G_N a^2}{\bar{\varphi}} \sum_l \left( \frac{\bar{p}_l}{\bar{\varphi}} \delta\varphi - \delta p_l \right), \tag{62}$$

where we have of course also split $\varphi$ into a background contribution and a perturbation, $\varphi = \bar{\varphi} + \delta\varphi$. The equation coming from the part that is proportional to $\hat{k}_i \hat{k}_j$ of the Einstein $ij$ component reads,

$$h'' + 6\eta'' + 2\mathcal{H}(h' + 6\eta') - 2\eta k^2 = -\frac{24\pi G_N}{\bar{\varphi}} a^2 (\bar{\rho} + \bar{p})\sigma. \tag{63}$$

The zero and spatial component of the covariant conservation equation of the vacuum are, respectively,

$$0 = \frac{\bar{\varphi}'}{\varphi} \left( \frac{\delta\rho}{\bar{\rho}} - \frac{\delta\varphi}{\bar{\varphi}} \right) + \frac{\delta\varphi'}{\bar{\varphi}} - \frac{\delta\rho'_{\text{vac}}}{\bar{\rho}}, \tag{64}$$

$$0 = \bar{\varphi}'(\bar{p} + \bar{\rho})\theta - k^2(\bar{p}\delta\varphi + \bar{\varphi}\delta\rho_{\text{vac}}), \tag{65}$$

where in the last two equations

$$\delta\rho = \sum_l \delta\rho_l \, ; \qquad \bar{p} = \sum_l \bar{p}_l \, ; \qquad \bar{\rho} = \sum_l \bar{\rho}_l \, ; \qquad (\bar{p} + \bar{\rho})\theta = \sum_l (\bar{p}_l + \bar{\rho}_l)\theta_l \equiv g. \tag{66}$$

It is possible to isolate $\delta\rho_{\text{vac}}$ from (65) and substitute it into the other equations. The only difficulty is found when the substitution is performed in (64), since in order to do this we need to evaluate the quantity $g' \equiv [(\bar{p} + \bar{\rho})\theta]'$. CLASS computes $g = [(\bar{p} + \bar{\rho})\theta]$ but not its derivative. If we knew how to write $[(\bar{p} + \bar{\rho})\theta]'$ in terms of $\delta\varphi$, $\delta\varphi'$, $g$, and other accessible quantities, then we could obtain the differential equation for $\delta\varphi$ from (64) and implement it in CLASS without defining explicitly $\delta\rho_{\text{vac}}$, just incorporating the effect of the vacuum perturbation directly into the equations. This is actually possible. Let us start by differentiating (65). We obtain:

$$\delta\rho'_{\text{vac}} = -\frac{\bar{p}}{\bar{\varphi}}\delta\varphi' - \frac{\bar{p}'}{\bar{\varphi}}\delta\varphi + \frac{\bar{\varphi}''g + \bar{\varphi}'g' - (\bar{\varphi}')^2 g/\bar{\varphi}}{k^2\bar{\varphi}} + \frac{\bar{\varphi}'\bar{p}\delta\varphi}{\bar{\varphi}^2}. \tag{67}$$

Note that the quantity $g'$, which we have defined previously, appears in this expression. As CLASS does not provide it to us we need to evaluate it by adding some supplementary piece in the code. It is possible to obtain $g'$ upon differentiating (61) and combining it with (62). The result reads

$$g' = \frac{k^2 \bar{\varphi}\mathcal{H}}{12\pi G_N a^2}(h' - k^2 f') + k^2\left(\delta p - \frac{\bar{p}\delta\varphi}{\bar{\varphi}}\right) + \left(\frac{\bar{\varphi}'}{\bar{\varphi}} - 2\mathcal{H}\right)g, \tag{68}$$

where $f \equiv (h + 6\xi)/k^2$ and its derivative, $f'$, are quantities that we can obtain from CLASS. In addition, $h'$ is provided by CLASS. Now, we only need to substitute $g'$ from (68) into (67), and substitute the resulting expression for $\delta\rho'_{\text{vac}}$ into (64). In doing so, we finally obtain the equation for the $\delta\varphi$ perturbation:

$$\delta\varphi' = \frac{\bar{\rho}}{\bar{\rho} + \bar{p}}\left[\delta\varphi\left(\frac{\bar{\varphi}'}{\bar{\varphi}} - \frac{\bar{p}'}{\bar{\rho}} + \frac{\bar{\varphi}'}{\bar{\varphi}}\frac{\bar{p}}{\bar{\rho}}\right) - \bar{\varphi}'\frac{\delta\rho}{\bar{\rho}} + \frac{\bar{\varphi}''g + \bar{\varphi}'g' - g(\bar{\varphi}')^2/\bar{\varphi}}{k^2\bar{\rho}}\right]. \tag{69}$$

The initial condition for $\delta\varphi$ is easy to find. Deep in the RDE we can neglect all the terms proportional to $\bar{\varphi}'$ and $\bar{\varphi}''$, so we are left with the following simple equation:

$$\delta\varphi' = -\delta\varphi\frac{\bar{p}'}{\bar{\rho} + \bar{p}} \longrightarrow a\frac{d\delta\varphi}{da} = \delta\varphi \longrightarrow \delta\varphi = \tilde{C}a, \tag{70}$$

where in the RDE we have used $\bar{p} = \frac{1}{3}\bar{\rho} = \frac{1}{3}\rho_r^0 a^{-4}$, of course. The background value of $\varphi$, i.e., $\bar{\varphi}$, remains almost constant during the RDE. It only grows very mildly with $a$, with a constant of proportionality which is small. Hence, we would expect the constant $\tilde{C}$ to be small as well. In any case, we expect $\delta\varphi \approx 0$ at $z_{\text{ini}} = 10^{14}$, which is the initial condition that we use for $\delta\varphi$ in our modified version of CLASS.

## 5. Data and Methodology

We fit the $\Lambda$CDM model, the running vacuum models under consideration (the type-I RRVM, the type-I RRVM$_{\text{thr.}}$ and the type-II RRVM), and finally the XCDM [150] (also called $w$CDM), a generic parameterization of the dynamical DE whose dark energy EoS, $w_0$, is constant and is one parameter of the fit (expected to lie near $-1$). To test the response of the XCDM along with the relevant models under consideration can be useful, as it serves as a benchmark scenario for generic models of dynamical dark energy. We fit all these models to a large, robust, and updated set of cosmological observations from all the main sources. Our dataset involves observations from: (i) distant type Ia supernovae (SNIa); (ii) baryonic acoustic oscillations (BAO); (iii) a compilation of (differential age) measurements of the Hubble parameter at different redshifts ($H(z_i)$); (iv) large-scale structure (LSS) formation data (specifically, an updated list of data points on the observable $f(z_i)\sigma_8(z_i)$); and, finally, (v) CMB Planck 2018 data of different sorts. A brief description now follows of each of these datasets along with the corresponding references.

**SNIa**: We consider the data from the so-called 'Pantheon+' compilation [151], which contains the apparent magnitudes and redshifts associated with 1701 light curves obtained from 1550 SNIa in the redshift range $0.001 \leq z \leq 2.26$. See Section 2.2 of [152] for details of the theoretical formulae employed to take into account these data points. Interestingly, the new Pantheon+ compilation also includes the 77 light curves from the 42 SNIa in the host galaxies employed by the SH0ES team in their analysis [153,154]. The distance to the host galaxies has been measured with calibrated Cepheids. The inclusion of these luminosity distances in our dataset will be made clear by adding the label "+SH0ES". They break the existing full degeneracy between $H_0$ and the absolute magnitude of SNIa, $M$, when only SNIa are considered in the analysis. The SH0ES calibration of the supernovae in conjunction with the cosmic distance ladder leads to larger preferred values of the Hubble parameter of $73.04 \pm 1.04$ km/s/Mpc [153]. This large value, as compared to Planck's measurement ($67.36 \pm 0.54$ km/s/Mpc, obtained from the TT,TE,EE+lowE+lensing data [155]), is at the root of the $\sim 5\sigma$ $H_0$ tension.

**BAO**: We employ 13 data points on isotropic and anisotropic BAO estimators. See Table 1 for the exact values and the corresponding references.

**Cosmic chronometers**: In our analyses, we use 32 data points on the Hubble parameter $H(z_i)$ measured with the differential age technique [156]. They span the redshift range $0.07 \leq z \leq 1.965$. We provide the complete list of data points and the corresponding references in Table 2. We have considered the effect of the known correlations between the various data points, as explained in [131]; see also Table 2 and its caption. The covariance matrix has been computed using the script provided in the following link[6]

**LSS**: Fifteen large-scale structure (LSS) data points between $0.01 \lesssim z \lesssim 1.5$, embodied in the observable $f(z_i)\sigma_8(z_i)$, which is known as the weighted linear growth rate, with $f(z)$ being the so-called growth factor and $\sigma_8(z)$ the root-mean-square mass fluctuations on the $R_8 = 8h^{-1}$ Mpc scale. See Table 3 for the complete list of data points and the corresponding references. We can take advantage of the relation $f(z)\sigma_8(z) = -(1+z)\frac{d\sigma_8(z)}{dz}$ to compute this quantity. The function $\sigma_8(z)$ involves the matter power spectrum, which is computed numerically by our modified version of the Einstein–Boltzmann code CLASS. It is important to note that this way of computing $f(z)\sigma_8(z)$ can only be used provided that we are in the linear regime, since in this case, and in our models, the matter density contrast can be written as $\delta_m(a,k) = D(a)F(k)$, where the dependence on the scale factor and the comoving wave number $k$ is factored out. The term $D(a)$ is known as the growth function and $F(k)$ encodes the initial conditions[7].

**CMB**: For the cosmic microwave background data, we utilize the full Planck 2018 TT,TE,EE+lowE likelihood [15]. This incorporates the information of the CMB temperature and polarization power spectra, and their cross-correlation. We refer to this dataset simply as "CMB". We also test separately the Planck 2018 TT+lowE likelihood, which does not include the effect of the high-$\ell$ multipoles of the CMB polarization spectrum, in order to check the impact of this particular dataset on our fitting results. It is also useful to compare with our previous analyses [85], in which only this type of CMB data were used. In our fitting scenarios, we indicate the removal of the high-$\ell$ CMB polarization data from the complete CMB likelihood with the label "CMB (No pol.)".

As described in the preceding lines, for the SNIa we may or may not include the information provided by the SH0ES team, whereas for the CMB we can consider the effect of the high-$\ell$ polarization data or not. An alternative calibration method of the absolute magnitude of SNIa based on the tip of the red giant branch [159,160] instead of Cepheids yields a measurement of $H_0$ somewhat in the middle of those provided by Planck [15] and SH0ES [153], $H_0 = 69.8 \pm 1.7$ km/s/Mpc[8]. In addition, it is also convenient to test the impact of the CMB polarization data from Planck, since previous works in the literature have found a moderate inconsistency between them and the Planck CMB temperature data, both in the 2015 [40] and 2018 [42] releases. This inconsistency could be due to a deficiency of the $\Lambda$CDM or the presence of unaccounted systematics in the data. Hence, these arguments motivate us to explore the following four different datasets.

- **Baseline**: In our Baseline dataset, we consider the string SNIa+BAO+$H(z)$+LSS+CMB. Note that here we do not include the SH0ES data.
- **Baseline+SH0ES**: The Baseline dataset is in this case complemented with the apparent magnitudes of the SNIa in the host galaxies and their distance moduli employed by SH0ES.
- **Baseline (No pol.)**: The same as in the Baseline case, but now removing the high-$\ell$ polarization data from the CMB likelihood. That is to say, we have replaced the "CMB" dataset with "CMB (No pol.)".
- **Baseline (No pol.)+SH0ES**: The same as in "Baseline (No pol.)", but including also the data from SH0ES.

These are the four datasets that we employ to constrain our models. We present our fitting results in Tables 4–7, Figures 1–3, and also in Tables A1–A4 of Appendix A.

**Table 4.** Mean values with 68% confidence intervals obtained from our fitting analysis of our Baseline dataset, composed by the string SNIa+BAO+$H(z)$+LSS+CMB. We display the values of the different cosmological parameters: the Hubble parameter ($H_0$), the reduced baryon and CDM density parameters ($\omega_b \equiv \Omega_b^0 h^2$ and $\omega_{cdm} \equiv \Omega_{cdm}^0 h^2$, respectively, with $\Omega_i^0 \equiv 8\pi G_N \rho_i^0 / 3H_0^2$), the current nonrelativistic matter density parameter ($\Omega_m^0$), the equation of state of the vacuum/DE fluid ($w_0$), the effective parameter of the running vacuum ($\nu_{eff}$) (see (22) and (55)), the initial and current values of the variable $\varphi \equiv G_N/G$, the optical depth to reionization ($\tau_{reio}$), the amplitude and spectral index of the primordial power spectrum ($A_s$ and $n_s$, respectively), the absolute magnitude of SNIa ($M$), the rms mass fluctuations at $8h^{-1}$ Mpc scale at present time ($\sigma_8$), the derived parameter $S_8 \equiv \sigma_8 \sqrt{\Omega_m^0/0.3}$, and the comoving sound horizon at the drag epoch ($r_d$). We also show the incremental value of DIC with respect to the $\Lambda$CDM, denoted $\Delta$DIC.

| | Baseline | | | | |
|---|---|---|---|---|---|
| Parameter | $\Lambda$CDM | Type-I RRVM | Type-I RRVM$_{thr.}$ | Type-II RRVM | XCDM |
| $H_0$(km/s/Mpc) | $68.27 \pm 0.35$ | $68.22 \pm 0.47$ | $67.65 \pm 0.38$ | $68.12 \pm 0.97$ | $67.49 \pm 0.56$ |
| $\omega_b$ | $0.02251 \pm 0.00013$ | $0.02253 \pm 0.00015$ | $0.02252 \pm 0.00013$ | $0.02247 \pm 0.00020$ | $0.02258 \pm 0.00013$ |
| $\omega_{cdm}$ | $0.11803 \pm 0.00078$ | $0.11807 \pm 0.00078$ | $0.1248 \pm 0.0019$ | $0.1181 \pm 0.0011$ | $0.11712 \pm 0.00094$ |

**Table 4.** *Cont.*

| | | | Baseline | | |
|---|---|---|---|---|---|
| Parameter | **ΛCDM** | **Type-I RRVM** | **Type-I RRVM$_{\text{thr.}}$** | **Type-II RRVM** | **XCDM** |
| $\Omega_{\text{m}}^{0}$ | $0.3029 \pm 0.0045$ | $0.3036 \pm 0.0056$ | $0.3235 \pm 0.0071$ | $0.3032 \pm 0.0089$ | $0.3082 \pm 0.0055$ |
| $w_0$ | $-1$ | $-1$ | $-1$ | $-1$ | $-0.962 \pm 0.022$ |
| $\nu_{\text{eff}}$ | - | $0.00006 \pm 0.00030$ | $0.0227 \pm 0.0055$ | $-0.00008 \pm 0.00035$ | - |
| $\varphi_{\text{ini}}$ | - | - | - | $1.006 \pm 0.024$ | - |
| $\varphi(0)$ | - | - | - | $1.008 \pm 0.028$ | - |
| $\tau_{\text{reio}}$ | $0.0512 \pm 0.0073$ | $0.0511 \pm 0.0080$ | $0.0601 \pm 0.0082$ | $0.0505 \pm 0.0078$ | $0.0546 \pm 0.0077$ |
| $\ln(10^{10}A_{\text{s}})$ | $3.033 \pm 0.015$ | $3.032 \pm 0.016$ | $3.053 \pm 0.017$ | $3.031 \pm 0.016$ | $3.038 \pm 0.016$ |
| $n_{\text{s}}$ | $0.9698 \pm 0.0035$ | $0.9701 \pm 0.0038$ | $0.9707 \pm 0.0035$ | $0.9681 \pm 0.0069$ | $0.9722 \pm 0.0038$ |
| $M$ | $-19.415 \pm 0.010$ | $-19.416 \pm 0.014$ | $-19.429 \pm 0.011$ | $-19.420 \pm 0.030$ | $-19.432 \pm 0.014$ |
| $\sigma_8$ | $0.8003 \pm 0.0064$ | $0.799 \pm 0.011$ | $0.7733 \pm 0.0092$ | $0.801 \pm 0.010$ | $0.7885 \pm 0.0093$ |
| $S_8$ | $0.804 \pm 0.010$ | $0.803 \pm 0.011$ | $0.803 \pm 0.010$ | $0.805 \pm 0.015$ | $0.802 \pm 0.011$ |
| $r_{\text{d}}$ (Mpc) | $147.46 \pm 0.21$ | $147.47 \pm 0.25$ | $147.44 \pm 0.21$ | $147.9 \pm 1.9$ | $147.62 \pm 0.23$ |
| $\Delta DIC$ | - | $-2.04$ | $+15.34$ | $-4.18$ | $+1.74$ |

In order to study the performance of the various models when they are confronted with the wealth of cosmological data, we define the joint $\chi^2$-function as follows,

$$\chi_{\text{tot}}^2 = \chi_{\text{SNIa}}^2 + \chi_{\text{BAO}}^2 + \chi_H^2 + \chi_{\text{LSS}}^2 + \chi_{\text{CMB}}, \tag{71}$$

where $\chi_{\text{CMB}}^2$ and $\chi_{\text{SNIa}}^2$ may include or not the contribution of the high-$\ell$ CMB polarization and SH0ES data, respectively, depending on the dataset that we consider.

To solve the background and perturbation equations of the type-I RRVM, type-I RRVM$_{\text{thr.}}$, and type-II RRVM we make use of our own modified versions of the Einstein–Boltzmann system solver `CLASS` [102,103], which is now equipped with the additional features that we have briefly described in the previous sections. We explore and put constraints on the parameter spaces of our models with Markov chain Monte Carlo (MCMC) analyses. More specifically, we make use of the Metropolis–Hastings algorithm [166,167], which is already implemented in the Monte Carlo sampler `MontePython`[9] [168,169]. We stop the MCMC when the Gelman–Rubin convergence statistic is $R - 1 < 0.02$ [170,171], and analyze the converged chains with the Python code `GetDist`[10] [172] to compute the mean values of the cosmological parameters, their confidence intervals, and the posterior distributions.

We have set conservative flat priors for the input parameters in the MCMC, much wider than their marginalized posterior distributions. For the six primary cosmological parameters that are common in all the models, we use: $0.005 < \omega_{\text{b}} < 0.1$, $0.001 < \omega_{\text{cdm}} < 0.99$, $20 < H_0, [\text{km/s/Mpc}] < 100$, $1.61 < \ln(10^{10}A_s) < 3.91$, $0.8 < n_s < 1.2$, and $0.01 < \tau_{\text{reio}} < 0.8$. The type-I RRVM and type-I RRVM$_{\text{thr.}}$ have one additional degree of freedom (*d.o.f.*) compared to the ΛCDM, which is encoded in the parameter $\nu$. We use the flat prior $-0.5 < \nu_{\text{eff}} < 0.5$. On the other hand, the type-II RRVM is characterized by two extra parameters, $\nu_{\text{eff}}$ and the initial value of $\varphi$, for which we use the priors $-\frac{1}{6} < \nu_{\text{eff}} < \frac{1}{6}$ and $0.7 < \varphi_{\text{ini}} < 1.3$. Finally, for the constant dark energy EoS parameter of the XCDM model we employ the prior $-3 < w_0 < 0.2$. In all our analyses we set the current temperature of the CMB to $T_0 = 2.7255$ K [173], and consider three neutrino species, approximated as two massless states and a massive neutrino of mass $m_\nu = 0.06$ eV.

To compare the fitting performance of the various models under study from a Bayesian perspective, we utilize the deviance information criterion (DIC) [174], which takes into account the presence of extra *d.o.f.'s* by duly penalizing the inclusion of additional parame-

ters in the model; see, e.g., the review [175] for a summarized discussion on how to use and interpret the information criteria in the cosmological context. The DIC value can be computed through the following equation:

$$\text{DIC} = \chi^2(\bar{\theta}) + 2p_D. \tag{72}$$

In this equation, $p_D = \overline{\chi^2} - \chi^2(\bar{\theta})$ represents the effective number of parameters and $2p_D$ is the so-called 'model complexity'. The latter is the quantity employed in the DIC criterion to penalize the presence of extra *d.o.f.'s*. The term $\overline{\chi^2}$ is the mean value of the $\chi^2$-function, which is obtained from the Markov chains. In this sense the computation of the DIC is a more sophisticated procedure of model comparison than other information criteria such as the Akaike information criterion [176]. Finally, $\bar{\theta}$ in (72) represents the mean value of the fitting parameters.

**Table 5.** Same as in Table 4, but adding the information from SH0ES to our Baseline dataset.

| | Baseline +SH0ES | | | | |
|---|---|---|---|---|---|
| **Parameter** | **ΛCDM** | **Type-I RRVM** | **Type-I RRVM$_{\text{thr.}}$** | **Type-II RRVM** | **XCDM** |
| $H_0$(km/s/Mpc) | $68.82 \pm 0.33$ | $69.17 \pm 0.43$ | $68.33 \pm 0.35$ | $70.79 \pm 0.69$ | $68.67 \pm 0.50$ |
| $\omega_b$ | $0.02264 \pm 0.00013$ | $0.02253 \pm 0.00015$ | $0.02266 \pm 0.00013$ | $0.02281 \pm 0.00017$ | $0.02265 \pm 0.00013$ |
| $\omega_{\text{cdm}}$ | $0.11697 \pm 0.00073$ | $0.11685 \pm 0.00075$ | $0.01227 \pm 0.0018$ | $0.1178 \pm 0.0011$ | $0.11679 \pm 0.00089$ |
| $\Omega_m^0$ | $0.2961 \pm 0.0041$ | $0.2928 \pm 0.0049$ | $0.3128 \pm 0.0064$ | $0.2808 \pm 0.0058$ | $0.2971 \pm 0.0047$ |
| $w_0$ | $-1$ | $-1$ | $-1$ | $-1$ | $-0.993 \pm 0.020$ |
| $\nu_{\text{eff}}$ | - | $-0.00037 \pm 0.00029$ | $0.0197 \pm 0.0055$ | $-0.00003 \pm 0.00033$ | - |
| $\varphi_{\text{ini}}$ | - | - | - | $0.949 \pm 0.016$ | - |
| $\varphi(0)$ | - | - | - | $0.950 \pm 0.021$ | - |
| $\tau_{\text{reio}}$ | $0.0533 \pm 0.0074$ | $0.0501 \pm 0.0079$ | $0.0617^{+0.0081}_{-0.0095}$ | $0.0523 \pm 0.0077$ | $0.0539 \pm 0.0078$ |
| $\ln(10^{10}A_s)$ | $3.035 \pm 0.015$ | $3.031 \pm 0.016$ | $3.053^{+0.017}_{-0.019}$ | $3.041 \pm 0.016$ | $3.036 \pm 0.016$ |
| $n_s$ | $0.9726 \pm 0.0035$ | $0.9705 \pm 0.0037$ | $0.9736 \pm 0.0034$ | $0.9824 \pm 0.0058$ | $0.9730 \pm 0.0037$ |
| $M$ | $-19.3989 \pm 0.0096$ | $-19.390 \pm 0.012$ | $-19.410 \pm 0.010$ | $-19.339 \pm 0.021$ | $-19.402 \pm 0.012$ |
| $\sigma_8$ | $0.7978 \pm 0.0064$ | $0.808 \pm 0.011$ | $0.7747 \pm 0.0093$ | $0.807 \pm 0.010$ | $0.7955 \pm 0.0089$ |
| $S_8$ | $0.7927 \pm 0.0094$ | $0.799 \pm 0.011$ | $0.7910 \pm 0.0098$ | $0.781 \pm 0.013$ | $0.801 \pm 0.010$ |
| $r_d$ (Mpc) | $147.59 \pm 0.21$ | $147.44 \pm 0.25$ | $147.60 \pm 0.21$ | $143.3 \pm 1.4$ | $147.63 \pm 0.23$ |
| ΔDIC | - | $-0.64$ | $+10.94$ | $+6.58$ | $-1.92$ |

Given a model *X*, we define the DIC difference with respect to the vanilla model (or concordance ΛCDM) in a way such that a positive difference of DIC implies that the new model (*X*) fares better than the vanilla model (and hence that *X* provides smaller values of DIC than the concordance model), whereas negative differences mean just the opposite, that is, that model *X* fares worse than the vanilla model. Therefore, the appropriate definition is

$$\Delta\text{DIC} \equiv \text{DIC}_{\Lambda\text{CDM}} - \text{DIC}_X. \tag{73}$$

In our case, X represents either the type-I or type-II running vacuum models in their RRVM implementation; and also the XCDM, which, as indicated before, is used as a benchmark scenario for dynamical DE. In the usual argot of the information criteria, values $0 \leq \Delta\text{DIC} < 2$ are said to entail *weak* evidence in favor of the considered option beyond the standard model. However, if $2 \leq \Delta\text{DIC} < 6$, one then speaks of *positive* evidence, whilst if $6 \leq \Delta\text{DIC} < 10$, it is considered that there is *strong* evidence in favor of the non-standard model *X*. Finally, if it turns out that $\Delta\text{DIC} > 10$, one may licitly claim (according to the rules of these information criteria) that there is *very strong* evidence supporting the model under

study as compared to the vanilla cosmology. In contrast, if the statistical parameter (73) proves negative, it is an unmistakable sign that the vanilla cosmology is favored over model *X* by the observational data.

**Table 6.** Same as in Table 4, but without including the high-$\ell$ CMB polarization data from Planck in our combined dataset.

| | Baseline (No pol.) | | | | |
|---|---|---|---|---|---|
| Parameter | $\Lambda$CDM | Type-I RRVM | Type-I RRVM$_{thr.}$ | Type-II RRVM | XCDM |
| $H_0$(km/s/Mpc) | $68.29 \pm 0.38$ | $68.10 \pm 0.48$ | $67.66 \pm 0.41$ | $68.8 \pm 1.2$ | $67.31 \pm 0.56$ |
| $\omega_b$ | $0.02228 \pm 0.00019$ | $0.02235 \pm 0.00022$ | $0.02231 \pm 0.00019$ | $0.02242 \pm 0.00026$ | $0.02239 \pm 0.00020$ |
| $\omega_{cdm}$ | $0.11746 \pm 0.00085$ | $0.11744 \pm 0.00086$ | $0.1242 \pm 0.0019$ | $0.1166 \pm 0.0016$ | $0.1160 \pm 0.0011$ |
| $\Omega_m^0$ | $0.3011 \pm 0.0048$ | $0.3029 \pm 0.0056$ | $0.3215 \pm 0.0072$ | $0.294 \pm 0.011$ | $0.3068 \pm 0.0055$ |
| $w_0$ | $-1$ | $-1$ | $-1$ | $-1$ | $-0.948 \pm 0.022$ |
| $\nu_{eff}$ | - | $0.00025 \pm 0.00038$ | $0.0223 \pm 0.0056$ | $0.00028 \pm 0.00043$ | - |
| $\varphi_{ini}$ | - | - | - | $0.982 \pm 0.030$ | - |
| $\varphi(0)$ | - | - | - | $0.976 \pm 0.035$ | - |
| $\tau_{reio}$ | $0.0489^{+0.0084}_{-0.0076}$ | $0.0508 \pm 0.0083$ | $0.0581 \pm 0.0082$ | $0.0508 \pm 0.0083$ | $0.0540 \pm 0.0080$ |
| $\ln(10^{10}A_s)$ | $3.026^{+0.018}_{-0.016}$ | $3.028 \pm 0.016$ | $3.047 \pm 0.017$ | $3.030 \pm 0.017$ | $3.034 \pm 0.016$ |
| $n_s$ | $0.9695 \pm 0.0037$ | $0.9712 \pm 0.0045$ | $0.9703 \pm 0.0037$ | $0.9754 \pm 0.0087$ | $0.9736 \pm 0.0041$ |
| $M$ | $-19.415 \pm 0.0011$ | $-19.420 \pm 0.014$ | $-19.429 \pm 0.012$ | $-19.397 \pm 0.038$ | $-19.436 \pm 0.0014$ |
| $\sigma_8$ | $0.7965 \pm 0.0069$ | $0.790 \pm 0.013$ | $0.7710 \pm 0.0094$ | $0.792 \pm 0.012$ | $0.7799 \pm 0.0098$ |
| $S_8$ | $0.798 \pm 0.011$ | $0.793 \pm 0.013$ | $0.798 \pm 0.011$ | $0.783 \pm 0.020$ | $0.793 \pm 0.012$ |
| $r_d$ (Mpc) | $147.86 \pm 0.30$ | $148.00 \pm 0.35$ | $147.81 \pm 0.30$ | $146.5 \pm 2.4$ | $148.15 \pm 0.33$ |
| $\Delta$DIC | - | $-1.84$ | $+14.54$ | $-3.06$ | $+3.82$ |

**Table 7.** Same as in Table 4, but removing the high-$\ell$ polarization data from Planck and including the information provided by SH0ES.

| | Baseline (No pol.) +SH0ES | | | | |
|---|---|---|---|---|---|
| Parameter | $\Lambda$CDM | Type-I RRVM | Type-I RRVM$_{thr.}$ | Type-II RRVM | XCDM |
| $H_0$(km/s/Mpc) | $68.94 \pm 0.37$ | $69.10 \pm 0.44$ | $68.48 \pm 0.39$ | $71.69 \pm 0.80$ | $68.61 \pm 0.51$ |
| $\omega_b$ | $0.02247 \pm 0.00018$ | $0.02240 \pm 0.00022$ | $0.02251 \pm 0.00018$ | $0.02280 \pm 0.00024$ | $0.02252 \pm 0.00019$ |
| $\omega_{dm}$ | $0.11630 \pm 0.00083$ | $0.11632 \pm 0.00083$ | $0.1220 \pm 0.0019$ | $0.1160 \pm 0.0015$ | $0.1157 \pm 0.0010$ |
| $\Omega_m^0$ | $0.2933 \pm 0.0045$ | $0.2919 \pm 0.0062$ | $0.3095 \pm 0.0067$ | $0.2702 \pm 0.0068$ | $0.2950 \pm 0.0048$ |
| $w_0$ | $-1$ | $-1$ | $-1$ | $-1$ | $-0.981 \pm 0.021$ |
| $\nu_{eff}$ | - | $-0.00022 \pm 0.00036$ | $0.0193 \pm 0.0055$ | $0.00048 \pm 0.00040$ | - |
| $\varphi_{ini}$ | - | - | - | $0.919^{+0.019}_{-0.022}$ | - |
| $\varphi(0)$ | - | - | - | $0.908^{+0.025}_{-0.028}$ | - |
| $\tau_{reio}$ | $0.0512 \pm 0.0074$ | $0.0494 \pm 0.0084$ | $0.0595^{+0.0082}_{-0.0092}$ | $0.0528 \pm 0.0085$ | $0.0533 \pm 0.0079$ |
| $\ln(10^{10}A_s)$ | $3.029 \pm 0.016$ | $3.027 \pm 0.017$ | $3.047^{+0.017}_{-0.019}$ | $3.041 \pm 0.017$ | $3.032 \pm 0.016$ |
| $n_s$ | $0.9728 \pm 0.0036$ | $0.9715 \pm 0.0044$ | $0.9739 \pm 0.0037$ | $0.9915 \pm 0.0070$ | $0.9744 \pm 0.0041$ |
| $M$ | $-19.396 \pm 0.011$ | $-19.392 \pm 0013$ | $-19.406 \pm 0.011$ | $-19.311 \pm 0.024$ | $-19.403 \pm 0.013$ |
| $\sigma_8$ | $0.7939 \pm 0.0068$ | $0.801 \pm 0.014$ | $0.7719 \pm 0.0094$ | $0.794 \pm 0.012$ | $0.7876 \pm 0.0096$ |

**Table 7.** *Cont.*

| | Baseline (No pol.) +SH0ES | | | | |
|---|---|---|---|---|---|
| Parameter | ΛCDM | Type-I RRVM | Type-I RRVM$_{\text{thr.}}$ | Type-II RRVM | XCDM |
| $S_8$ | $0.785 \pm 0.010$ | $0.790 \pm 0.014$ | $0.784 \pm 0.010$ | $0.754 \pm 0.017$ | $0.781 \pm 0.011$ |
| $r_{\text{d}}$ (Mpc) | $147.97 \pm 0.30$ | $147.85 \pm 0.85$ | $147.92 \pm 0.30$ | $141.3 \pm 1.6$ | $148.08 \pm 0.32$ |
| ΔDIC | - | $-0.10$ | $+10.06$ | $+13.78$ | $-0.96$ |

## 6. Discussion of the Results

The class of running vacuum models (RVMs) has proven to be theoretically sound and thus worth being studied phenomenologically. It emerges as a generic framework out of renormalizable QFT in curved spacetime; in fact, one which is capable of describing the expansion history of the universe from the early times to our days from first principles [17,19,20]. If we take the quantum vacuum seriously, the RVM framework is a natural consequence of it. The predicted changes are not dramatic, but can be crucial to fit the pieces together. Indeed, the phenomenological expectations from the running vacuum approach on the cosmological observables remain always very close to the ΛCDM, as can be seen from the fitting results displayed in Tables 4–7. Nevertheless, small departures are definitely predicted owing to the presence of vacuum fluctuations from the quantized matter fields in the FLRW background. These vacuum effects must be properly renormalized in the QFT context, and as a result they bring about small "radiative corrections" on top of the standard ΛCDM predictions—recall their generic form in Equation (1). They have been computed in detail in [29–32] and can help to fix the phenomenological hitches currently besetting the standard model of cosmology, which is strictly based on GR and no quantum effects at all. The generic RVM contains a few free parameters amenable to fitting from the cosmological data, but the formal structure of the quantum effects is unambiguous and well defined. In fact, the quantum corrections at low energy appear to be proportional to $\sim H^2$ and $\sim \dot{H}$, as shown in Equation (1). These corrections induce a dynamics in the physical value of the VED and the corresponding physical value of the cosmological term, $\Lambda$. In other words, in the RVM these quantities acquire a cosmological evolution rather than remaining strictly constant as in the ΛCDM. This fact may have phenomenological consequences worth studying. In the present work, we have dwelled upon particular realizations of the RVM exhibiting a rich phenomenology and we have studied the conditions by which they may offer a helping hand to curb one or both tensions ($\sigma_8$ and $H_0$) under study.

In this section, we discuss in detail the results we have obtained for particular RVM realizations, which in all cases are sourced by the same formal QFT structure indicated in Equation (1), and compare them with those obtained with the ΛCDM and the popular XCDM parameterization of the dark energy EoS parameter [150]. Above all, we should remark at this point that the results obtained here are fully consistent with those reported in our last study confronting the RVMs against the overall cosmological observations [85]. In the present instance, however, we have updated our datasets and have extended significantly the reach of our considerations by displaying a much more comprehensive numerical study; see Tables 4–7 and Figures 1–4. Most significantly, the current presentation includes for the first time the effect of the CMB polarization data from Planck. In fact, we recall that the companion analysis of [85] focused exclusively on the Planck 2018 TT+lowE data, and hence was not sensitive to the influence from the high-$\ell$ polarizations. In contrast, in the current study we use the two full likelihoods from Planck, namely, Planck 2018 TT+lowE and Planck 2018 TT,TE,EE+lowE (cf. Section 5) and compare their distinct impact on the fitting results of each of the RVM realizations under focus, viz. the type-I and type-II implementations.

Let us start with the results obtained with the RRVM of type I, first, under the assumption that the vacuum interacts with dark matter during the entire cosmic history.

The CMB data from Planck put very tight constraints on the amount of dark energy at the decoupling time (see, e.g., [177]) and, therefore, on the RVM parameter that controls the exchange of energy in the dark sector, $\nu_{eff}$. We obtain central values of $\nu_{eff} \sim \mathcal{O}(10^{-4})$ with all our datasets, with associated error bars that make our measurements compatible with 0 at $\lesssim 1\sigma$ c.l., indicating no statistical preference for a non-null vacuum dynamics in the universe in the context of this model. Although the type-I RRVM is fundamentally different from the $\Lambda$CDM, its phenomenology is in practice quite similar, due to the strong upper bounds on $\nu_{eff}$. This explains why the constraints obtained on the other cosmological parameters are so similar in the two models, and also the small impact the type-I RRVM has on the cosmological tensions. This is the conclusion that follows if we assume that the cosmological solution that we have found for the type-I models is valid all the way from the present time up to the point in the radiation-dominated epoch where we have placed our initial conditions following the standard setup of CLASS (see, however, below). We refer the reader to Tables 4–7 for the detailed list of the fitting results. In particular, we would like to mention that the results quoted in the last two tables (namely, Tables 6 and 7, where the CMB data are used without polarizations) are perfectly compatible within error bars (both in order of magnitude and sign) with the results obtained in our previous analysis [85].

Nevertheless, we cannot exclude the possibility that the vacuum dynamics undergoes a transition between two (or more) epochs of the expansion history, e.g., through a change in the value of $\nu_{eff}$ or of the effective EoS parameter $w_{vac}$. As previously noted, the possibility of a tomographic behavior of the DE throughout the cosmic expansion has been explored previously in the literature; see, e.g., [107–110]. In the RVM case, we are further motivated to think of a scenario of this sort since it is actually suggested within the context of the QFT calculation supporting the RVM structure; see [31].

While we shall not go into theoretical details here, we have opted for mimicking such a (continuous, although quite abrupt) transition with a phenomenological $\Theta$-function approach. Thus, we have explored the simplest scenario (with just one transition) in the context of what we have called the type-I RRVM$_{thr.}$, i.e., the type-I model with a threshold. We thereby assume that the interaction between vacuum and dark matter is activated only at a threshold redshift lower than $z_* = 1$[11]. We have chosen this transition redshift after performing a fitting analysis allowing $z_*$ to vary freely in the Monte Carlo process. The value $z_* \sim 1$ turns out to maximize the posterior. When the Baseline dataset is employed, we find $\nu_{eff} = 0.0227 \pm 0.0055$ and, hence, significant evidence for a late-time vacuum decay into dark matter at $4.1\sigma$ c.l. This allows the suppression of the clustering in the universe at $z < z_*$, as is clear from the fitting value $\sigma_8 = 0.773 \pm 0.009$ reported in Table 4 and the left-most plots in Figure 1. The small values of matter fluctuations at linear scales allows us to essentially solve the tension with the $f\sigma_8$ data (see Figure 4 and [45]), decreasing the value of $\chi^2_{f\sigma_8}$ by $\sim 9$ units with respect to the standard model, while keeping the good description of the other datasets (cf. Table A1).

This is very remarkable and completely aligned with our previous results [85], in which we already showed the outstanding capability of our model for producing lesser growth in the late universe[12]. The vacuum decay leads to a decrease in the VED and an enhancement of $\rho_m$ at present. This produces larger values of the current $\Omega_m$, which somehow compensates for the decrease in $\sigma_8$ and gives rise to values of $S_8(=\tilde{S}_8)$ of the same order as those obtained in the $\Lambda$CDM and the other models studied in this paper. The comparison of the results for the type-I RRVM$_{thr.}$ reported in our Tables 4–7 also demonstrates the robustness and stability of the fitting output under changes in the dataset. The values of DIC obtained with the Baseline configuration, and also considering the SH0ES data with and without the use of the CMB polarization information from Planck, are in the range $10 \lesssim \text{DIC} \lesssim 15$. Therefore, we find in all cases *very strong* evidence for this model from a Bayesian perspective, i.e., after penalizing the use of the extra parameter $\nu_{eff}$. The smallest values of DIC$\sim 10$ are obtained when the SH0ES data are also used in the analysis. This triggers a decrease in the evidence for non-zero vacuum dynamics, which still renders

at the $\sim 3.5\sigma$ c.l. The model is able to solve the $\sigma_8$ tension, but does not alleviate the Hubble tension, since the values of $H_0$ stay close to those found in the $\Lambda$CDM.

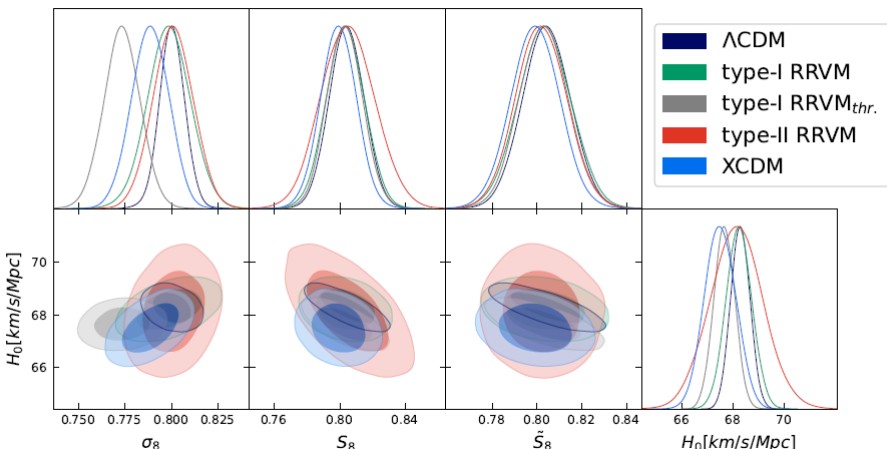

**Figure 1.** Contour plots at $1\sigma$ and $2\sigma$ c.l. in the $\sigma_8 - H_0$, $S_8 - H_0$, and $\tilde{S}_8 - H_0$ planes and their corresponding one-dimensional posteriors, obtained from the fit of the various models to the Baseline dataset (cf. Section 5). The parameter $\tilde{S}_8 \equiv S_8/\sqrt{\varphi(0)}$ can only differ from the standard $S_8$ in the type-II RRVM; see the main text of Section 6 and [85,119]. The type-I RRVM$_{thr.}$ can explain a value of $\sigma_8 \sim 0.78$, much smaller than in the other models. This is accompanied by a $4.1\sigma$ evidence for a non-zero value of the RVM parameter $\nu_{\text{eff}}$; see Table 4. We find in all cases similar values of $\tilde{S}_8$ and $H_0$ to those found in $\Lambda$CDM, but the type-II RRVM has a much wider posterior for this parameter, and hence this model can accommodate a larger Hubble constant. See also the comments in the main text.

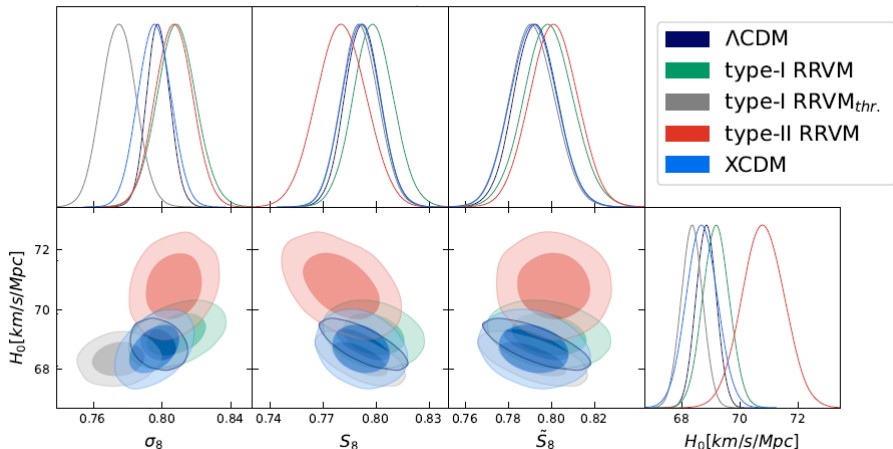

**Figure 2.** Same as in Figure 1, but using the Baseline+SH0ES dataset (cf. Section 5). The inclusion of the data from SH0ES shifts the one-dimensional posterior of $H_0$ towards $H_0 \sim 71$ km/s/Mpc in the type-II RRVM, a region that is still allowed by the Baseline dataset, cf. Figure 1. Remarkably, the small values of $\sigma_8$ found in the type-I RRVM$_{thr.}$ remain stable, and no important differences between the models are found regarding the value of $\tilde{S}_8$. The lower value of $S_8$ obtained in the type-II RRVM is due to the fact that this parameter does not account for the $2.4\sigma$ departure of $\varphi(0)$ from 1, $\varphi(0) = 0.950 \pm 0.021$; see the caption of Table 4.

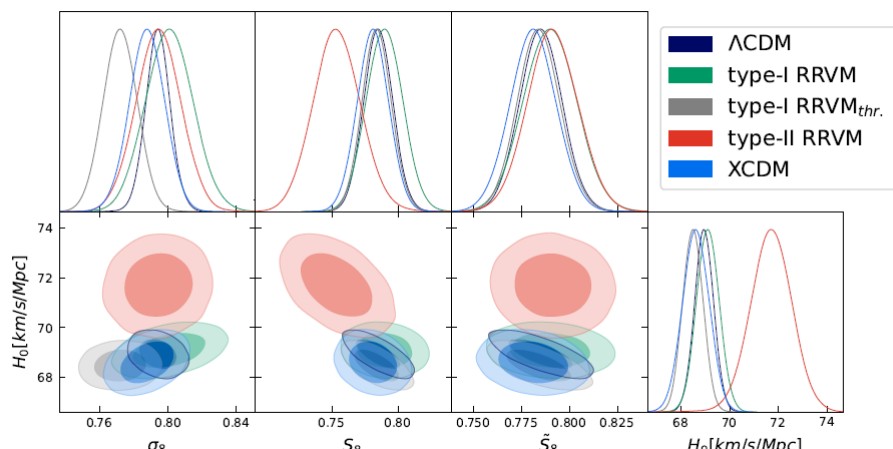

**Figure 3.** Same as in Figure 2, but removing the high-$\ell$ polarization data from Planck, i.e., considering the Baseline (No pol.)+SH0ES dataset (cf. Section 5). Again, as in the other fitting analyses, the value of $\sigma_8$ is kept small in the type-I RRVM$_{\text{thr.}}$. The absence of CMB polarization data allows for even smaller values of $\varphi(0)$ in the type-II RRVM, $\varphi(0) = 0.908^{+0.025}_{-0.028}$, which is now $3.5\sigma$ below the GR value $\varphi(0) = 1$. This explains the large value of $H_0 \sim 72$ km/s/Mpc, which basically renders the Hubble tension insignificant, below the $1\sigma$ c.l.

Let us now move on to the type-II RRVM. Here, we switch off the exchange of energy between the vacuum and the (dark) matter sector, but in compensation give allowance for a possible variation in $G$ at cosmological scales, which is induced by the running of the vacuum in accordance with the Bianchi identity. In this type of model, therefore, the current value of the gravitational coupling may depart from $G_N$. We have previously defined the auxiliary variable $\varphi(z) \equiv G_N/G(z)$ to parameterize such a departure. Let us also note that type-II models have two additional free parameters as compared to the $\Lambda$CDM (one more than type-I models): $\nu_{\text{eff}}$ and $\varphi_{\text{ini}}$ (the latter being the initial value of $\varphi$ at high redshift in the radiation-dominated epoch, before it starts evolving very slowly with the cosmic evolution). Since $G$ can vary for these models, stringent constraints on type-II models should apply from the existing limits on the relative variation in the gravitational coupling if one assumes that the cosmological value of $G$ must satisfy them (cf. Section 4.1). These constraints are in fact satisfied by our fitting results for this type of running vacuum model. In fact, regardless of the dataset we use to fit the model, we obtain values of $\nu_{\text{eff}} \sim \mathcal{O}(10^{-4})$ compatible with 0. There is no clear hint of vacuum dynamics in this case. However, in the limit $\nu_{\text{eff}} \to 0$ we recover the $\Lambda$CDM only if $\varphi(0) \to 1$. This is actually the crucial ingredient that can make the type-II model a rather appealing framework for relieving the $H_0$ tension, but only in the presence of the SH0ES data, as we now explain. In its absence, we obtain values of $\varphi(0)$ compatible with 1 (the strict GR value) at $< 1\sigma$ c.l. Using the Baseline dataset we find $\varphi(0) = 1.008 \pm 0.028$, whereas we find $\varphi(0) = 0.982 \pm 0.030$ with the Baseline (No pol.) alternative, which as we know is the same set but excluding the polarizations. It is obvious that the polarization data favor larger values of $\varphi$ (closer to 1) or, equivalently, smaller values of $G$ (closer to $G_N$)[13]. The improvement in the description of the data compared to the $\Lambda$CDM is in both cases only marginal within the Baseline scenario, with or without polarizations (cf. Tables A1 and A3). This is indeed reflected in the negative $\Delta$DIC values gathered in both cases, viz. $\Delta$DIC$\sim -(3-4)$, which point to a *positive* preference for the standard cosmological model (see Tables 4 and 6). Now, in stark contrast with the meager situation just described with the Baseline dataset, the inclusion of the SH0ES data produces a dramatic turnaround of the results in the desired direction. It shifts the posterior of $\varphi(0)$ towards a region of lower values, which is more prominent in the absence of the CMB polarization likelihoods, to wit: $\varphi(0) = 0.950 \pm 0.021$ in the Baseline+SH0ES analysis and $\varphi(0) = 0.908^{+0.025}_{-0.028}$ in the Baseline (No pol.)+SH0ES one. This

produces a significant decrease in the comoving sound horizon at the baryon-drag epoch $r_d$, which now lies in the ballpark $r_s \sim 141 - 143$ Mpc rather than in the usual higher range $147 - 148$ Mpc usually preferred by the $\Lambda$CDM model. This fact, together with a significant increase in the spectral index of the primordial power spectrum $n_s > 0.98$ [119], generates a noticeable increase in $H_0$, whose fitting constraints in the context of our analysis now read $H_0 = 70.8 \pm 0.7$ km/s/Mpc and $H_0 = 71.7 \pm 0.8$ km/s/Mpc, respectively[14]. The upshot is that the Hubble tension is basically washed out in this running vacuum model scenario[15]. The incremental DIC values with respect to the vanilla model corroborate in fact *strong*, or even *very strong*, evidence in favor of running vacuum, depending on whether the CMB polarization data are considered or not. As remarked, this happens only when we include the information from SH0ES and at the expense of worsening a little the description of the CMB temperature spectrum—cf. the supplementary Tables A2 and A4 in Appendix A, in which we display the breakdown of the different $\chi^2$ contributions from each observable. Regarding the description of the LSS, the type-II RRVM is not able to improve the fit to the $f\sigma_8$ data with respect to the other models under study, as is clear from Figure 4 and the tables in Appendix A. The model allows the posterior values of $S_8$ to be shifted towards the region preferred by the weak lensing measurements, more conspicuously in the analysis with the Baseline (No pol.)+SH0ES dataset. However, $S_8$ might not be the most correct quantity to make contact with observations in models with a renormalized gravitational coupling at cosmological scales. Alternative estimators, such as $\tilde{S}_8$ (see the captions of Figures 1–3), might be more appropriate. The values of $\tilde{S}_8$ are larger than those of $S_8$ and similar to those found in the other models explored in this work, including the $\Lambda$CDM.

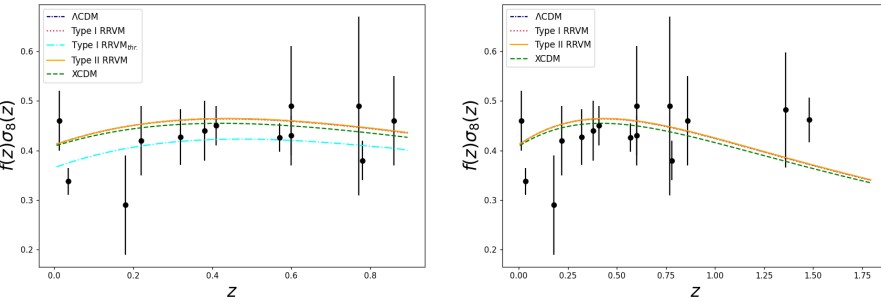

**Figure 4.** Theoretical curves of $f(z)\sigma_8(z)$ for the various models together with the observational data points listed in Table 3. We have employed the central values of the Baseline fitting analysis (cf. Table 4). The type-I RRVM$_{\text{thr}}$ has the ability to solve the $\sigma_8$ tension by suppressing the clustering at $z < 1$.

Finally, we comment on the results obtained with the generic XCDM parameterization. It is well known that a quintessence EoS parameter $w_0 > -1$ allows the suppression of the amount of structure in the universe due to the increase in dark energy in the past, which fights against the aggregation of matter. It is also known, though, that quintessence cannot alleviate the Hubble tension because the decaying nature of the DE makes in this case the critical density and, hence, also $H_0$ to be smaller at low redshifts; see, e.g., [86,183,184]. If we do not employ the SH0ES data, i.e., if we use the Baseline and Baseline (No pol.) datasets, we find a $\sim 2\sigma$ deviation of the EoS parameter from $w_0 = -1$ (a pure $\Lambda$) in the quintessence region. This is in accordance with what we have already mentioned. The LSS data, which points to a lower level of clustering than in the $\Lambda$CDM, prefers quintessence. Actually, we obtain small values of $\sigma_8 \sim 0.78$. The XCDM is able to relieve the tension with the LSS data in the absence of SH0ES, but the decrease in $\chi^2_{f\sigma_8}$ is not as big as for the type-I RRVM$_{\text{thr.}}$; see, e.g., Table A1. One can see in that table that the contribution to $\chi^2_{f\sigma_8}$ is significantly less (roughly a factor of two smaller) for the type-I model with threshold than in a generic XCDM parameterization. On the other hand, the improvement is less

robust for the XCDM, this being corroborated by the fact that the hints of DE dynamics disappear when we include the data from SH0ES, since the latter favors a phantom dark energy EoS parameter. This shifts $w_0$ towards smaller values. For instance, in the analysis with Baseline+SH0ES we obtain $w_0 = -0.993 \pm 0.020$. The phantom region, however, is not attained because we include LSS data in our analysis. In fact, the structure formation data do not favor the phantom region since in that case the amount of DE is smaller in the past, and this does not help to prevent the excess of structure formation, which is tantamount to saying that it does not help to relieve the $\sigma_8$ tension. Thus, a compromise is needed, and in the presence of the SH0ES data the XCDM provides a value of the EoS closer to $w = -1$ than in the absence of such data. If LSS data were not used, the SH0ES data would succeed in pushing the EoS of the XCDM to the phantom domain [177]. In contrast to this voluble behavior of the EoS for a generic DE fluid, the type-I $\mathrm{RRVM_{thr.}}$ provides a substantially better overall fit and its effective DE behavior is quintessence-like in the structure formation region up to the present day. Indeed, we find $\nu_{\mathrm{eff}} > 0$ (both with or without SH0ES data) at a large confidence level of $(3.5 - 4)\sigma$. Hence, the vacuum energy density associated with that model is indeed decreasing with the expansion within the relevant region of structure formation for both datasets, Baseline or Baseline+SH0ES, with or without polarizations, cf. Tables 4–7.

## 7. Conclusions

In this work, we have put to the test a class of dynamical dark energy (DDE) models that go under the name of running vacuum models (RVMs). These have been discussed for a long time in the literature; see, e.g., [17,19,20] as well as [185,186] and references therein. These models have successfully withstood a number of litmus tests against all types of modern data, thus demonstrating their maturity and robustness as serious competitors to the concordance ΛCDM model, this being true not only in regard to their fitting power but also, and indeed especially, in regard to improving the status of the ΛCDM and generalizations thereof in the context of theoretical physics. The essential new feature of the RVM class is that it predicts the existence of DDE associated with the vacuum, a fundamental concept in QFT. Put another way, the running vacuum shows up here as if it were a form of DDE, but in truth is (quantum) vacuum after all—not just another artifact extracted from the blackbox of the DE aimed at mimicking or supplanting the fundamental notion of vacuum energy in QFT. In the RVM paradigm, there is no rigid cosmological term, Λ, owing to the fundamental need for renormalization of the VED in QFT. The scale of renormalization is dynamical and hence the computed quantum corrections produce a time-evolving VED with the expansion [20]. The general structure of the RVM has been recently buttressed by explicit calculations in the context of QFT in curved spacetime. We should mention that the smooth VED dynamics in the RVM was long suspected from semi-qualitative renormalization group arguments, see the aforesaid references and corresponding bibliography, but this was only recently substantiated in a fully fledged QFT context; see the detailed works [29–31]. Within the RVM, the gravitational coupling, $G$, will also be running in general. From its dynamical interplay with the vacuum energy density (VED), $\rho_{\mathrm{vac}}(H)$, we find that $G$ evolves very mildly as a logarithmic function of the Hubble rate, $G = G(\ln H)$. As it turns out, what we call Λ (as a physical quantity, not just as a formal parameter) in the RVM formulation, is actually nothing but the nearly sustained value of $8\pi G(H)\rho_{\mathrm{vac}}(H)$ around (any) given epoch. There is no such thing as a true cosmological constant in the RVM framework, and as a matter of fact it is fair to say that a (physically measurable) rigid parameter of this sort is not to be expected in renormalizable QFT [20].

As for the specific details of the phenomenological analysis put forward in this work, and for the sake of a better comparison with previous studies—particularly with the most recent one in [85]—in the current presentation we have focused on an implementation of the RVM which we have denoted RRVM. It has one single (extra) parameter in the type-I formulation, $\nu_{\mathrm{eff}}$, and two additional free parameters ($\nu_{\mathrm{eff}}$ and $\varphi_{\mathrm{ini}}$) in the type-II

RRVM, with respect to the $\Lambda$CDM. The VED has a dynamical component proportional to the scalar of curvature, $\mathcal{R}$, with $\nu_{\text{eff}}$ being its coefficient; see Equation (6). Such a coefficient can be accounted for analytically in QFT (see the above mentioned works) but it depends on the masses of all the quantized matter fields, so in practice it must be fitted to the overall cosmological data. This is actually the main task that we have undertaken in the present work. In so doing, we have found significant evidence that the VED is running with the cosmic expansion. In fact, upon performing a global fit to the cosmological observations from a wealth of data sources of all the main sorts, thus involving the full string SNIa+BAO+$H(z)$+LSS+CMB of relevant cosmological observables, and comparing the rigid option $\nu_{\text{eff}} = 0$ (namely, $\Lambda$ =const. corresponding to the $\Lambda$CDM model), with the running vacuum one ($\nu_{\text{eff}} \neq 0$), we find that a mild dynamics of the cosmic vacuum ($\nu_{\text{eff}} \sim 10^{-4} - 10^{-2}$) is highly favored, depending on the model. For type-I RRVM with threshold we find *very strong direct* evidence of such vacuum dynamics through a nonvanishing value of $\nu_{\text{eff}}$ at more than $4\sigma$ c.l. and an overall statistical score of $\Delta$DIC$> +10$ with respect to the vanilla model, whereas for type-II RRVM the evidence is also *strong*, but indirect, through the change in $G$, which leads to a favorable scenario when we consider the SH0ES data at the level of $\Delta$DIC$> +6(+10)$, depending on whether we use CMB polarization data or not. We have also checked that the improvement of the fit is not just caused by a generic form of the DDE, meaning that when we test if a simple XCDM ($w$CDM) parameterization [150] would do a similar job we meet a negative result, i.e., in the latter case we do not observe any significant amelioration with respect to the $\Lambda$CDM fit.

This is in stark contrast to the fitting results from the running vacuum. As previously indicated, for type-I models the level of evidence turns out to be *very strongly* supported by the DIC criterion (according to the conventional parlance used within the information criteria), provided there exists a threshold redshift for the DDE near the present day ($z_* \simeq 1$) where the vacuum evolution becomes suddenly activated in the RRVM form. For the sake of simplicity, here we have mimicked it just through a $\Theta$-function. With a mild level of dynamics, as indicated above, the $\sigma_8$ tension is rendered essentially nonexistent. The relief of the $H_0$ tension, on the other hand, can be significantly accomplished only within the type-II model with variable $G$ (the tension subsisting only at an inconspicuous level of $< 2\sigma$). Finally, let us note that even though the type-I model cannot deal with the $H_0$ tension, the overall fit quality that it offers in the presence of a DDE threshold is really outstanding. Specifically, the DIC difference with respect to the vanilla $\Lambda$CDM is $\Delta$DIC$= +15.34$, cf. Table 4. The type-I model with a threshold suppresses completely the $\sigma_8$ tension and provides a determination of $\nu_{\text{eff}} \neq 0$ at a level of significance slightly more than $4\sigma$. We note that this intriguing result would stay even if the $H_0$ tension faded away or suddenly disappeared. If the data on $f\sigma_8$ are free from unaccounted systematic errors, our results suggest, first of all, that it is very likely that the DE is dynamical and that it takes the running vacuum form; and, second, that such a vacuum dynamics started relatively recently ($z \sim 1$). This fact could be motivated by the same calculations underpinning the general RVM structure of the vacuum energy in QFT. The potential significance of these considerations cannot be overemphasized and we will certainly return to them in future studies.

**Author Contributions:** All the authors have made significant contributions to the research presented in this work. All authors have read and agreed to the published version of the manuscript.

**Funding:** JSP and CMP are funded by projects PID2019-105614GB-C21 and FPA2016-76005-C2-1-P (MINE CO, Spain), 2021-SGR-249 (Generalitat de Catalunya), and CEX2019-000918-M (ICCUB). AGV is funded by the Istituto Nazionale di Fisica Nucleare (INFN) through the project of the InDark INFN Special Initiative: "Dark Energy and Modified Gravity Models in the light of Low-Redshift Observations" (n. 22425/2020). JdCP is supported by the Margarita Salas fellowship funded by the European Union (NextGenerationEU).

**Data Availability Statement:** The data employed in this work is publicly available and can be found in the references and links provided in Section 5.

**Acknowledgments:** The work of JSP is partially supported by the COST Association Action (European Cooperation in Science and Technology) CA18108 "Quantum Gravity Phenomenology in the Multimessenger Approach (QG-MM)". This article is also based upon work from COST Action CA21136—Addressing observational tensions in cosmology with systematics and fundamental physics (CosmoVerse).

**Conflicts of Interest:** The authors declare no conflict of interest.

## Appendix A. Additional Tables

In this Appendix, we present Tables A1–A4, with the individual contributions of each observable to the total $\chi^2$ for all the fitting analyses performed in this work, obtained from the mean values of the cosmological parameters. These results must be close to the true $\chi^2_{\min}$, since the underlying posteriors are, to a very good approximation, Gaussian. We prefer not to use the latter, since the minimum $\chi^2$ found by `MontePython` is not always very precise; see footnote 10 in [169].

**Table A1.** Detailed breakdown of the different $\chi^2$ contributions from each observable in our Baseline dataset with the cosmological parameters reported in Table 4. We call the contribution that contains the correlations between the BAO and LSS datasets BAO-$f\sigma_8$ (correl.) (see the references of Table 1), whereas the uncorrelated contributions are simply called BAO and $f\sigma_8$.

| Baseline | | | | | |
|---|---|---|---|---|---|
| Experiment | $\Lambda$CDM | Type-I RRVM | Type-I RRVM$_{\text{thr.}}$ | Type-II RRVM | XCDM |
| CMB | 2770.70 | 2771.04 | 2770.14 | 2770.48 | 2773.68 |
| SNIa | 1405.49 | 1405.39 | 1403.24 | 1405.64 | 1402.82 |
| $f\sigma_8$ | 17.15 | 16.92 | 8.29 | 17.14 | 15.08 |
| BAO-$f\sigma_8$ (correl.) | 19.96 | 19.92 | 14.54 | 19.92 | 17.91 |
| $H(z)$ | 13.16 | 13.18 | 13.33 | 13.30 | 13.33 |
| BAO | 10.94 | 10.97 | 10.70 | 10.91 | 10.65 |
| $\chi^2_{\text{total}}$ | 4237.40 | 4237.42 | 4220.24 | 4237.39 | 4233.48 |

**Table A2.** Same as Table A1 but for the Baseline+SH0ES dataset. We have employed the parameters from Table 5.

| Baseline+SH0ES | | | | | |
|---|---|---|---|---|---|
| Experiment | $\Lambda$CDM | Type-I RRVM | Type-I RRVM$_{\text{thr.}}$ | Type-II RRVM | XCDM |
| CMB | 2774.02 | 2771.46 | 2774.02 | 2777.50 | 2774.72 |
| SNIa | 1490.46 | 1488.22 | 1491.62 | 1474.38 | 1490.65 |
| $f\sigma_8$ | 15.33 | 16.92 | 8.21 | 17.27 | 14.92 |
| BAO-$f\sigma_8$ (correl.) | 19.82 | 21.68 | 13.35 | 20.48 | 19.21 |
| $H(z)$ | 12.97 | 12.85 | 13.07 | 12.47 | 13.00 |
| BAO | 10.77 | 10.77 | 10.43 | 10.76 | 10.69 |
| $\chi^2_{\text{total}}$ | 4323.38 | 4321.91 | 4310.71 | 4312.85 | 4323.19 |

**Table A3.** Same as Table A1 but for the Baseline (No pol.) dataset and making use of the parameters provided in Table 6.

| Experiment | $\Lambda$CDM | Type-I RRVM | Type-I RRVM$_{\text{thr.}}$ | Type-II RRVM | XCDM |
|---|---|---|---|---|---|
| | **Baseline (No pol.)** | | | | |
| CMB | 1184.03 | 1185.16 | 1183.39 | 1184.93 | 1186.79 |
| SNIa | 1405.84 | 1405.51 | 1403.41 | 1405.60 | 1402.65 |
| $f\sigma_8$ | 16.03 | 14.99 | 8.27 | 15.35 | 13.38 |
| BAO-$f\sigma_8$ (correl.) | 19.44 | 19.12 | 14.12 | 19.20 | 16.99 |
| $H(z)$ | 13.20 | 13.29 | 13.35 | 12.85 | 13.38 |
| BAO | 10.91 | 10.97 | 10.65 | 10.96 | 10.47 |
| $\chi^2_{\text{total}}$ | 2649.45 | 2649.04 | 2633.19 | 2648.89 | 2643.66 |

**Table A4.** Same as Table A1 but for the Baseline (No pol.)+SH0ES dataset. In this case we have used the parameters listed in Table 7.

| Experiment | $\Lambda$CDM | Type-I RRVM | Type-I RRVM$_{\text{thr.}}$ | Type-II RRVM | XCDM |
|---|---|---|---|---|---|
| | **Baseline (No pol.) + SH0ES** | | | | |
| CMB | 1186.28 | 1185.07 | 1186.32 | 1189.76 | 1187.75 |
| SNIa | 1490.20 | 1489.18 | 1490.80 | 1469.30 | 1490.24 |
| $f\sigma_8$ | 14.15 | 15.17 | 8.16 | 15.07 | 13.10 |
| BAO-$f\sigma_8$ (correl.) | 20.49 | 21.50 | 14.07 | 19.25 | 19.26 |
| $H(z)$ | 12.97 | 12.91 | 12.91 | 12.79 | 13.01 |
| BAO | 10.86 | 10.86 | 10.48 | 10.86 | 10.64 |
| $\chi^2_{\text{total}}$ | 2734.95 | 2734.69 | 2722.74 | 2717.20 | 2734.00 |

## Notes

[1]   https://lesgourg.github.io/class_public/class.html, accessed on 25 May 2023.

[2]   In practice this means that we have first fitted the value of $z_*$ as one more free parameter in our analysis. Subsequently, we have assumed that the threshold point remains fixed at that point; see also [107–110] for a binned/tomographic approach to the DE. In our case, we have just one threshold whose existence might be motivated by QFT calculations [30,31].

[3]   If (dark) matter is not conserved but $G$ remains constant, we retrieve of course our previous scenario (16). In general, we may expect a mixture of both situations, but we shall refrain from dealing with the general case since it would introduce extra parameters; see, however, [111,112] for additional discussions that can be relevant for studies on the possible variation in the fundamental constants of nature.

[4]   It should be clear that $\varphi$ is not a dynamical degree of freedom, in contradistinction to Brans–Dicke-type theories of gravitation [113], and therefore $\varphi$ does not mediate any sort of long-range interaction that should be subdued by screening mechanisms.

[5]   Let us emphasize that Equation (56) is valid only in the MDE, and we have also pointed out that $\varphi \to$ const. in the DE epoch. This means that $G$ becomes more and more rigid when it transits from the MDE to the DE epoch, and therefore the actual limits on $\nu_{\text{eff}}$ are weaker than those that we have roughly estimated. This works to our benefit of course. In fact, a detailed calculation would require computing $\varphi$ in the DE epoch, but it proves unnecessary once we have shown that even in the most unfavorable case (i.e., when $\varphi$ evolves more rapidly than it actually does in the DE epoch) the obtained limits on $\nu_{\text{eff}}$ are nonetheless preserved by our fits. Notice that type-I models are totally unaffected by these limits, since $G$ is in this case constant, so $\nu_{\text{eff}}$ can be, in principle, larger for them.

[6]   https://gitlab.com/mmoresco/CCcovariance/-/blob/master/examples/CC_covariance.ipynb, accessed on 25 May 2023.

[7]   While it is common to rescale the measured values of $f\sigma_8$ by a factor $\frac{H(z)D_A(z)}{\tilde{H}(z)\tilde{D}_A(z)}$ to account for the Alcock–Paczynski (AP) effect [157] (in which the tildes denote the quantities computed in the fiducial cosmology employed by the galaxy surveys), there does not not seem to exist a general consensus on the exact correction to apply; see, e.g., [158] and references therein. In this sense, the above formula should be considered as just a rough estimate. We have checked that the AP-rescaling introduces negligible

shifts in our fitting results, a conclusion that is well in accordance with previous analyses in the literature [38,39,158]. For this reason, we have opted to not include this correction in our work.

8    This region is also preferred by late-time dynamical DE models when fitted to a very wide variety of background data that are independent from the direct cosmic distance ladder and CMB, $H_0 = 69.8 \pm 1.3$ km/s/Mpc [161]. See [162–165] for measurements of $H_0$ more in accordance with SH0ES obtained also with the tip of the red giant branch method.

9    https://baudren.github.io/montepython.html, accessed on 25 May 2023.

10    https://getdist.readthedocs.io/en/latest/, accessed on 25 May 2023.

11    See Section 3.3 for the practical implementation.

12    Note that our work [85] is previous to [178], in which the authors propose a friction mechanism between CDM and DE to suppress the clustering at $z \lesssim 1$.

13    This is something we already noted in previous studies within the context of the Brans–Dicke model with a cosmological constant [118,119].

14    Noticeably, the central values of $r_d$, $H_0$, and the absolute magnitude of SNIa, $M$, obtained for the type-II RRVM when the CMB polarization data are excluded in the fitting analysis are in very good agreement with the model-independent measurements from low-$z$ data reported in [152], which are also independent from the main drivers of the $H_0$ tension. For the Hubble constant, these authors find $H_0 = 71.6 \pm 3.1$ km/s/Mpc. However, these measurements still have large uncertainties and cannot arbitrate the Hubble tension yet; see also [179].

15    A similar phenomenology is found in the context of some modified gravity theories with a mild time evolution of $G$ and a non-negligible shift of its value with respect to $G_N$ [118,119,180]. See also [181,182].

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
