# Peer review of "Running Vacuum in the Universe: Phenomenological Status in Light of the Latest Observations, and Its Impact on the σ8 and H0 Tensions"

_universe, doi:10.3390/universe9060262_

Round 1

Reviewer 1 Report

See the attached referee report

Author Response

See attached pdf containing the response to the two referees

Reviewer 2 Report

Dear editor,

    This is a very interesting work about running vacuum dark energy models in light of the famous tension problems in cosmology. It is well written and deserved to be published, but the following flaws are required to be addressed by the authors.

1. The authors claimed that the running vacuum model given in Eq. (1) is predicted in the QFT. So the question is that does the QFT give any prediction to the values of the model parameters c_0, \mu and \tilde{\mu}? Then one can give comparison to the model parameters derived from the theoretical prediction and cosmic observations. I hope the authors address this issue properly.

2. In fact all the f\sigma_8(z) data are obtained assuming a fiducial \LambdaCDM cosmology, thus the Alcock-Paczynski effect should be considered for data analysis consistency. This flaw should be amended before the publication.

Author Response

(The authors gave the same response as above.)
